# BACH family members regulate angiogenesis and lymphangiogenesis by modulating VEGFC expression

Batya Cohen[1], Hanoch Tempelhof[1], Tal Raz[2], Roni Oren[1], Julian Nicenboim[1], Filip Bochner[1], Ron Even[1], Adam Jelinski[1], Raya Eilam[3], Shifra Ben-Dor[4] [ID], Yoseph Adaddi[4], Ofra Golani[4], Shlomi Lazar[5], Karina Yaniv[1], Michal Neeman[1] [ID]

Angiogenesis and lymphangiogenesis are key processes during embryogenesis as well as under physiological and pathological conditions. Vascular endothelial growth factor C (VEGFC), the ligand for both VEGFR2 and VEGFR3, is a central lymphangiogenic regulator that also drives angiogenesis. Here, we report that members of the highly conserved BACH (BTB and CNC homology) family of transcription factors regulate VEGFC expression, through direct binding to its promoter. Accordingly, down-regulation of *bach2a* hinders blood vessel formation and impairs lymphatic sprouting in a Vegfc-dependent manner during zebrafish embryonic development. In contrast, BACH1 overexpression enhances intratumoral blood vessel density and peritumoral lymphatic vessel diameter in ovarian and lung mouse tumor models. The effects on the vascular compartment correlate spatially and temporally with BACH1 transcriptional regulation of VEGFC expression. Altogether, our results uncover a novel role for the BACH/VEGFC signaling axis in lymphatic formation during embryogenesis and cancer, providing a novel potential target for therapeutic interventions.

## Introduction

The blood and lymphatic networks are two evolutionarily conserved transport systems that provide complementary functions in the maintenance of tissue homeostasis. In particular, the formation of new blood and lymphatic vessels is a prerequisite for vertebrate embryonic and postnatal development. During embryogenesis, the blood circulatory system is first to emerge. As early as mouse embryonic day (E) 7.5, vasculogenesis, the formation of a primitive vascular plexus through proliferation, migration, and differentiation of endothelial cells (ECs), takes place. Later on, this plexus undergoes massive remodeling via angiogenesis, which involves also arteriovenous differentiation (Chung & Ferrara, 2011). At E10.5, a distinct subpopulation of ECs within the cardinal vein commits to the lymphatic lineage, buds off, and migrates to establish primitive lymph sacs, which through further sprouting, give rise to the peripheral lymphatic system (Oliver & Alitalo, 2005; Semo et al, 2016). During adult life, most blood and lymphatic vessels are quiescent, with the exception of female reproductive organs during the ovarian cycle and the placenta during pregnancy. Reactivation of angiogenesis and lymphangiogenesis, however, is a hallmark of pathological processes associated with wound healing, myocardial infarction, allograft rejection, chronic inflammation, tumor progression, and malignant cell dissemination (Oliver & Alitalo, 2005; Chung & Ferrara, 2011).

The VEGF family of growth factors and its receptors are central signaling pathways, controlling angiogenesis and lymphangiogenesis during development and adulthood. VEGFs exert their activity by binding to the tyrosine kinase receptors (vascular endothelial growth factor receptor-1) VEGFR-1, VEGFR-2, and VEGFR-3 expressed in blood and lymphatic endothelial cells (LECs) (Lohela et al, 2009). VEGFA, the ligand for VEGFR-1 and VEGFR-2, is best known for its role in angiogenesis, stimulating EC proliferation and migration and increasing vascular permeability. VEGFB, which also binds to VEGFR1, plays a role in cell survival and indirectly promotes VEGFA-induced angiogenesis (Lal et al, 2018).

Vascular endothelial growth factor C (VEGFC) is another critical player in VEGF signaling. This protein, which signals through the receptors VEGFR-3 and VEGFR-2, plays a key role across species, guiding both lymphatic and blood vasculature development and remodeling (Karkkainen et al, 2004; Kuchler et al, 2006; Yaniv et al, 2006; Lohela et al, 2009; Hogan et al, 2009a; Gore et al, 2011; Villefranc et al, 2013; Shin et al, 2016). VEGFC is expressed by endothelial and non-endothelial cells, thus acting via both autocrine and paracrine signaling (Covassin et al, 2006b; Kodama et al, 2008; Lohela et al, 2008; Khromova et al, 2012; Helker et al, 2013; Villefranc et al, 2013). Evidence for its necessity for proper lymph formation is revealed by the finding that ectopic expression of *Vegfc* in mouse ear or skin keratinocytes results in hyperplasia of lymphatic vessels. Similarly, mice and zebrafish lacking VEGFC fail to develop a lymphatic vasculature as initial sprouting and directed migration of lymphatic progenitors from the cardinal vein are arrested (Karkkainen et al, 2004; Kuchler et al, 2006; Yaniv et al, 2006; Villefranc et al, 2013; Shin et al, 2016). In addition to its predominant

---

[1]Department of Biological Regulation, Weizmann Institute of Science, Rehovot, Israel    [2]Koret School of Veterinary Medicine, The Hebrew University of Jerusalem, Rehovot, Israel    [3]Department of Veterinary Resources, Weizmann Institute of Science, Rehovot, Israel    [4]Life Sciences Core Facilities, Weizmann Institute of Science, Rehovot, Israel    [5]Department of Pharmacology, Israel Institute for Biological Research, Ness-Ziona, Israel

Correspondence: karina.yaniv@weizmann.ac.il; michal.neeman@weizmann.ac.il

role in lymphangiogenesis, VEGFC was shown to induce angiogenesis upon overexpression in the mouse cornea, skin, or ischemic hind limb. Moreover, coronary vessel development is VEGFC dependent, and hearts devoid of VEGFC reveal dramatic delay in the formation of subepicardial sprouts (Chen et al, 2014). Recently, it has been shown by in vitro studies that the let-7a/TGFBR3 axis regulates angiogenesis through transcriptional regulation of VEGFC (Wang et al, 2019).

VEGFC has a critical role not only during development but also during tumor progression. In various human cancers, enhanced expression of VEGFC and higher levels of VEGFC in serum are commonly associated with tumor aggressiveness and lymph-node metastasis (Su et al, 2007; Lohela et al, 2009). In esophageal carcinoma, for instance, angiogenesis is driven via the phosphoinositide-phospholipase C-$\varepsilon$ (PI-PLC$\varepsilon$)/NF-$\kappa$B signaling pathway by direct promotion of VEGFC transcription (Chen et al, 2019). In xenograft or transgenic tumor models, stimulation of lymphangiogenesis by VEGFC promotes malignant cell dissemination (Stacker et al, 2014). Furthermore, blockade of Vegfc expression in tumor cells by stably transfected small interfering RNAs reduces lymphangiogenesis and lymph node metastasis of murine mammary cancers. Similarly, soluble VEGFR-3 protein has been shown to inhibit VEGFC-induced tumor lymphangiogenesis and metastatic spread in a breast cancer mouse model (Wissmann & Detmar, 2006).

A series of environmental and cellular factors were shown to modulate VEGFC expression. For instance, IL-6 and IL-17 regulate VEGFC expression via the PI3K-Akt or extracellular-signal-regulated kinase (ERK) 1/2 pathways, whereas MicroRNA-1826 significantly down-regulates VEGFC expression in human bladder cancer (Chen et al, 2012). Vegfc mRNA levels are heightened in the adipose tissue of obese mice, pointing to adipocytes as a source of elevated VEGFC levels in obesity (Karaman et al, 2015). In adults, inflammation induces robust up-regulation of VEGFC expression by macrophages (Baluk et al, 2005). In addition, various hormones, transcription factors, metallothioneins, and microenvironmental stresses (e.g., hyperthermia, oxidative stress, and high salt) were shown to control VEGFC expression (Cohen et al, 2009; Machnik et al, 2009; Sapoznik et al, 2009; Chen et al, 2012; Schuermann et al, 2015; Gauvrit et al, 2018).

Here, we report that BACH transcription factors, which are known to be involved in various intracellular signaling pathways, modulate VEGFC expression during embryonic development and tumor progression, offering a better understanding of blood and lymph vessel formation during physiological and pathological conditions.

# Results

## Spatial and temporal expression of *bach2* during zebrafish development

To explore the molecular mechanisms governing VEGFC expression, we searched for genes that co-express with VEGFC in human prostate cancer datasets (Glinsky et al, 2004). We identified 434 genes, on which we further applied the Promoters of Clusters analysis (Tabach et al, 2007). This method screens the promoters of genes with shared biological function against a library of transcription factor–binding motifs and identifies those which are statistically overrepresented. Five factors passed the basic significance threshold, out of which only

two possessed binding sites within the VEGFC promoter. We then searched for evolutionarily conserved regions, which often represent potential DNA regulatory elements (Ovcharenko et al, 2004). Comparison of the VEGFC promoters of human, mouse, and zebrafish revealed two highly conserved BACH (broadcomplex-tramtrack-bric-a-brac [BTB] and cap'n'col-lar type of basic leucine zipper [CNC-bZip] homology)-binding sequences (Fig 1A).

The BACH family of transcription factors comprises two members, BACH1 and BACH2. In mammals, BACH1 is expressed ubiquitously and has been shown to act either as an activator or repressor of transcription and to be involved in oxidative stress, metabolism, cell transformation, neurodegenerative diseases, tumor expansion, and metastatic spread (Watari et al, 2008; Warnatz et al, 2011; Nakanome et al, 2013; Igarashi & Watanabe-Matsui, 2014; Lee et al, 2014; Zhou et al, 2016; Lee et al, 2019; Lignitto et al, 2019; Wiel et al, 2019). BACH2 is a transcriptional repressor crucial for the terminal differentiation and maturation of both T and B lymphocytes (Sidwell & Kallies, 2016), and its loss is associated with severe autoimmune diseases. BACH proteins are highly conserved in vertebrates, particularly in the functional (BTB and bZip) domains and the regions immediately surrounding them (Igarashi & Watanabe-Matsui, 2014). Zebrafish have four *bach* genes with homology to mammalian BACH: *bach1a*, *bach1b*, *bach2a*, and *bach2b* (Zhang et al, 2014; Fuse et al, 2015; Luo et al, 2016).

As a first step, we evaluated the expression pattern of the different *bach* transcripts in ECs isolated by FACS from *Tg(fli1:EGFP)$^{y1}$* (Lawson & Weinstein, 2002) zebrafish embryos (Covassin et al, 2006a). We found the transcripts of *bach2a* and *bach2b* to be more abundant than *bach1b*, whereas *bach1a* is barely detectable in the GFP-positive (GFP$^+$) cell population, at either 21–24 hours post-fertilization (hpf) or 3 days post-fertilization (dpf) (Fig 1B). Consequently, we focused our studies on *bach2a* and *bach2b*, as they are the major paralogs expressed in the endothelium. To assess the spatial and temporal expression patterns of *bach2a* and *bach2b* during zebrafish embryogenesis, we performed whole-mount in situ hybridization. Abundant expression of *bach2a* and *bach2b* was apparent in the somites and somite boundaries at 20 hpf and up to 48 hpf (Fig 1C). Furthermore, we detected strong enrichment of *bach2a* transcripts in myotomes and in several areas of the central nervous system, including the hindbrain, midbrain–hindbrain boundary, midbrain, and forebrain (Fig S1).

## *bach2a* is essential for developmental angiogenesis and lymphangiogenesis

We then assessed the contribution of BACH to angiogenesis and lymphangiogenesis during embryonic development and, specifically, investigated their putative role as regulators of *vegfc* expression and function during zebrafish vasculature formation. We used antisense morpholino oligonucleotides (MOs) to knockdown the expression of the two *bach2* paralogs and then surveyed the phenotypic changes at different stages of development. At 30 hpf, clear defects were observed in the primordial hindbrain channel (PHBC) in *bach2a* MO-injected *Tg(fli1:EGFP)$^{y1}$* zebrafish embryos (Fig 2A and B). Similar phenotypes were detected in *vegfc* morphants (Fig 2A and B), in line with previous reports demonstrating impaired PHBC formation following the blocking of Vegfc-activated Vegfr3 signaling (Covassin et al, 2006b; Hogan et al, 2009b; Villefranc et al, 2013; Schuermann et al, 2015; Shin et al, 2016). In contrast to *bach2a* down-regulation, no defects were detected in *Tg(fli1:EGFP)$^{y1}$*

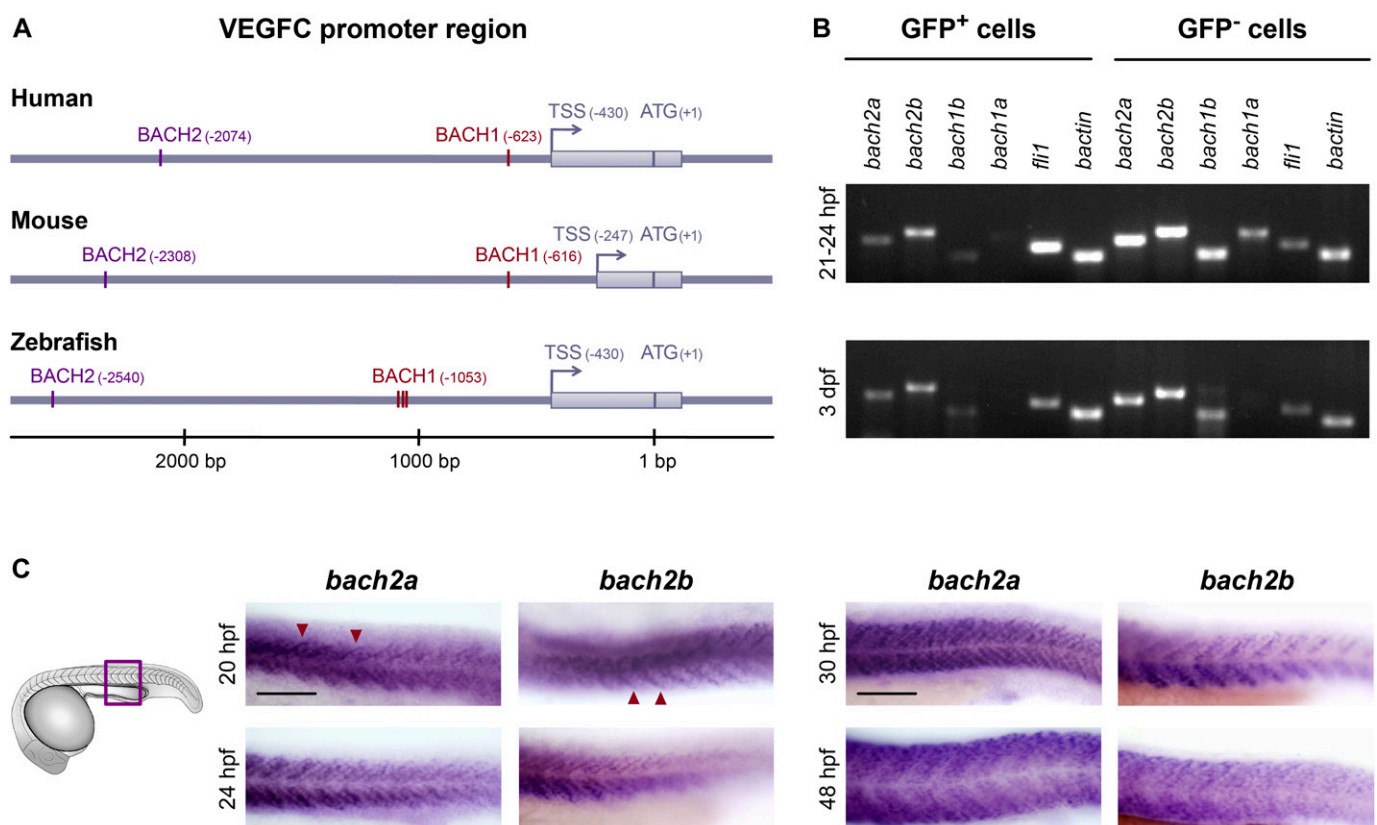

**Figure 1. Spatial and temporal expression of *bach2* paralog transcripts during zebrafish development.**
**(A)** BACH putative binding sites are evolutionarily conserved in the VEGFC promoter region. Numbering is from the ATG (translation initiation) because of the difference in length of the mouse's 5′ UTR. An arrow indicates transcription start site (TSS), and the location of the first exon is marked as a gray rectangle. The location of the BACH sites is as predicted by Genomatix Genome Analyzer MatInspector: Human (NM_005429.5 TSS at hg19, chr4:177713899 on the minus strand); Mouse (NM_009506.2 TSS at mm9, chr8:54077532 on the plus strand); and Zebrafish (NM_205734.1 TSS at Zv9, chr1:39270725 on the minus strand). **(B)** Semi-quantitative RT-PCR analysis of the indicated genes (*bactin-β* actin) in enriched GFP⁺ cells isolated by FACS from *Tg(fli1:EGFP)^y1* embryos at two developmental time points, 21–24 hpf and 3 dpf (two independent experiments for each time point). **(C)** A lateral view of the trunk region of a wild-type zebrafish embryo at 20, 24, 30, and 48 hpf, detected with a specific *bach2a* or *bach2b* anti-sense mRNA probe. A red arrowhead indicates somite boundaries. Scale bar, 100 μm.

embryos either injected with different concentrations of *bach2b* MOs (up to 10 ng per embryo, Fig 2A and C) or subjected to a *bach2b* CRISPR gRNA (*bach2b* gRNA, Figs S2A and B and 2A and B). The vascular abnormalities detected upon *bach2a* and *vegfc* knockdown were accompanied by pericardial and body edema, as well as reduced blood flow (Fig S3), resembling the phenotypes observed in collagen and calcium-binding EGF domain-1 (*ccbe1*) (Hogan et al, 2009a) and *vegfc* (Karkkainen et al, 2004; Shin et al, 2016) mutants. In contrast, no morphological abnormalities were observed after *bach2b* knockdown (Fig S4A–C).

To confirm the specificity of the *bach2a* morphant phenotype, we generated *bach2a* mutants using CRISPR/Cas9–mediated gene editing (Fig S2C–E). To our surprise, homozygous *bach2a* mutants displayed no PHBC defects (*bach2a^mut-/-*, Fig 2D and E). We, therefore, hypothesized that genetic compensation (Rossi et al, 2015; El-Brolosy & Stainier, 2017; El-Brolosy et al, 2019) through activation of *bach2b* could potentially account for the absence of angiogenic phenotypes in *bach2a* mutants. To address this possibility, we mated *bach2a^+/-* carriers and injected their progeny with either a sub-dose of *bach2b* MO or with *bach2b* gRNA that does not induce vascular malformations in WT embryos (Fig 2A–C). Interestingly, PHBC formation defects were identified in ~25% of the embryos, which upon genotyping were

found to carry the *bach2a* mutation (*bach2a^mut-/-* + *bach2b* MO and *bach2a^mut-/-* + *bach2b* gRNA, Fig 2D and E), as opposed to their wild-type siblings (*bach2a^mut+/+* + *bach2b* MO and *bach2a^mut+/+* + *bach2b* gRNA, Fig 2E). Conversely, after *bach2a* MO injection, *bach2a* wild-type siblings displayed PHBC formation defects, whereas homozygous mutants appeared normal (Fig 2D and E), suggesting that the *bach2a^mut-/-* mutants were less sensitive than their *bach2a^mut+/+* siblings to *bach2a* MO injections, and further confirming the specificity of both mutants and morphants. To ascertain whether the compensation mechanism involves up-regulation of the paralogous Bach2b gene, we assessed its expression in wild-type and *bach2a* mutants, by quantitative real-time-PCR. Surprisingly, although no significant differences in the levels of *bach2b* mRNA were detected (Fig S2F), we observed a slight elevation in *bach2a* mRNA levels in homozygous mutants (Fig S2G), suggesting a non-transcriptional mechanism. The underlying molecular mechanisms by which paralogous transcription factors compensate for each other's loss-of-function are relatively unexplored. A wide range of mechanisms may provoke robustness involving post-transcriptional or posttranslational regulation and pleiotropic effects. In addition, compensation may take place at the level of protein–protein interactions, whereby paralogs replace each other with respect

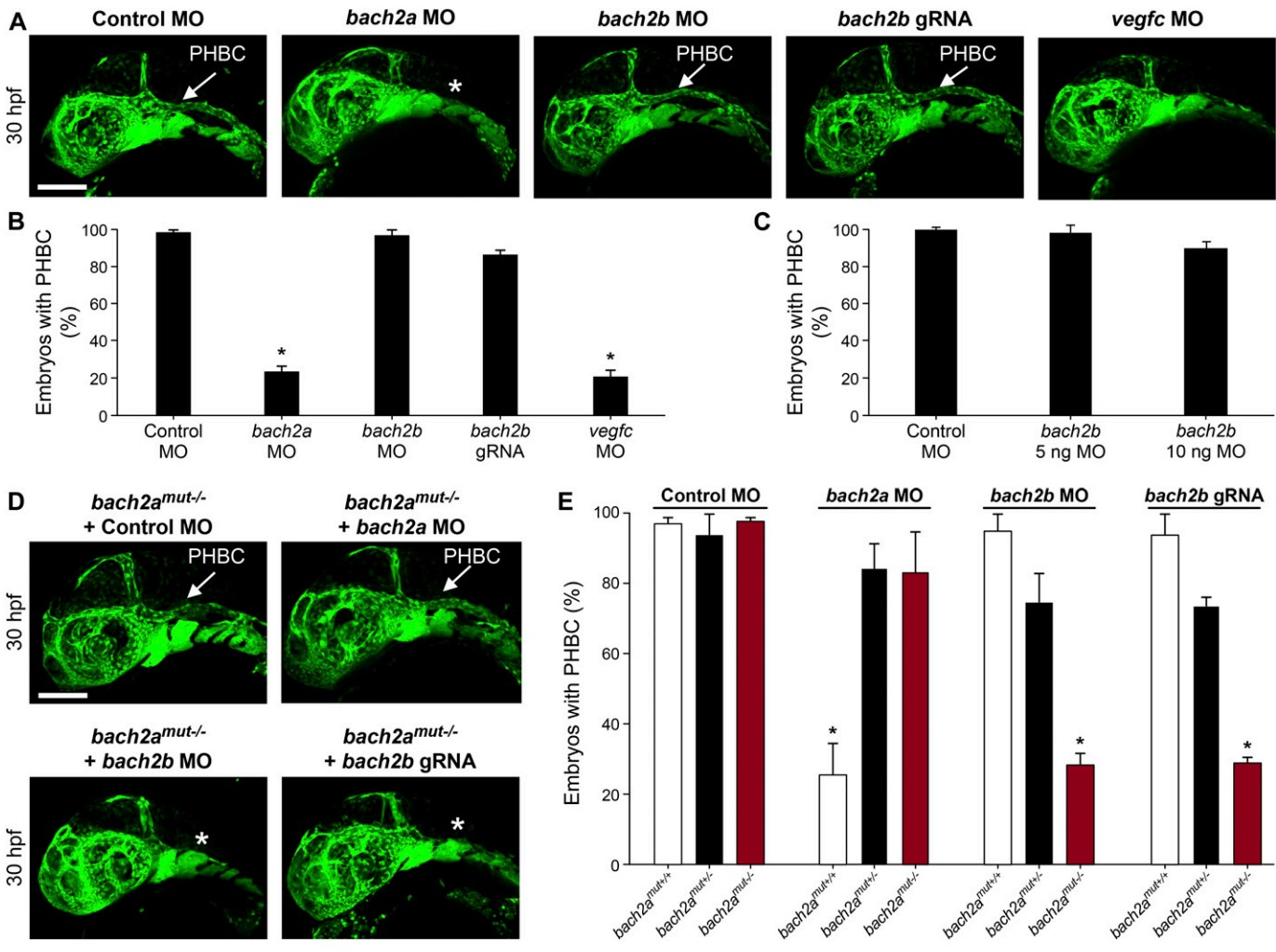

**Figure 2.   *bach2a* is essential for developmental angiogenesis in zebrafish embryos.**
**(A)** Confocal images of the primordial hindbrain channel (PHBC, white arrow) of 30-hpf *Tg(fli1:EGFP)$^{y1}$* embryos injected with control MO (10 ng), *bach2a* MO (3.75 ng), *bach2b* MO (3.75 ng), *bach2b* gRNA (125 ng), or *vegfc* MO (10 ng). Asterisk indicates the absence of PHBC. **(B)** Percentage of 30-hpf *Tg(fli1:EGFP)$^{y1}$* embryos with intact PHBC formation after injection with control MO (10 ng, $n_{Control\ MO}$ = 68), *bach2a* MO (3.75 ng, $n_{bach2a\ MO}$ = 107; *$P$ < 0.0001), *bach2b* MO (3.75 ng, $n_{bach2b\ MO}$ = 48), *bach2b* gRNA (125 ng, $n_{bach2b\ gRNA}$ = 42), or *vegfc* MO (10 ng, $n_{vegfc\ MO}$ = 35; *$P$ < 0.0001). Error bars, mean ± SEM. **(C)** Percentage of 30-hpf *Tg(fli1:EGFP)$^{y1}$* embryos with intact PHBC formation after injection with control MO (10 ng, $n_{Control\ MO}$ = 24) or an increased concentration of *bach2b* MO (5 ng, $n_{bach2b\ MO\ 5ng}$ = 24) or (10 ng, $n_{bach2b\ MO\ 10ng}$ = 24). Error bars, mean ± SEM; $P$ > 0.99999. **(D)** Confocal projection at 30 hpf *Tg(fli1:EGFP)$^{y1}$* of homozygous *bach2a* mutants (*bach2a$^{mut-/-}$*) from F2 *bach2a$^{mut+/-}$* incross. White arrow points at an intact PHBC detected in embryos injected with control MO (10 ng, *bach2a$^{mut-/-}$* + Control MO) and *bach2a* MO (3.75 ng, *bach2a$^{mut-/-}$* + *bach2a* MO). Asterisk indicates defects in PHBC development after injection with *bach2b* MO (3.75 ng, *bach2a$^{mut-/-}$* + *bach2b* MO) or *bach2b* gRNA (125 ng, *bach2a$^{mut-/-}$* + *bach2b* gRNA). Scale bar, 100 μm. **(E)** Percentage of randomly selected *bach2a$^{+/-}$* F2 incross progeny at 30 hpf with an intact PHBC formation injected with control MO (10 ng, $n_{bach2amut\ +\ Control\ MO}$ = 50; $P$ > 0.99999), *bach2a* MO (3.75 ng, $n_{bach2amut\ +\ bach2a\ MO}$ = 75; *$P$ < 0.0002), *bach2b* MO (3.75 ng, $n_{bach2amut\ +\ bach2a\ MO}$ = 55; *$P$ < 0.012) or *bach2b* gRNA (125 ng, $n_{bach2amut\ +\ bach2b\ gRNA}$ = 137; *$P$ < 0.001). After genotyping, offspring followed the expected Mendelian ratios of inheritance. Error bars, mean ± SEM. Kruskal–Wallis test in panels (B, C, E).

to their binding partners through ancestrally preserved binding ability. Thus, we speculate that other compensatory mechanisms (not transcriptional), yet unknown, may contribute to the compensation mechanism controlling *bach* robustness.

We then asked whether *bach2a* is also involved in lymphatic vessel development. To answer this question, we analyzed the effects of *bach2a* down-regulation on the formation of parachordal cells (PACs), the building blocks of the thoracic duct (TD), and the trunk lymphatic system in zebrafish (Yaniv et al, 2006; Nicenboim et al, 2015). A significantly reduced number of PAC-containing segments was detected in *bach2a* MO–injected *Tg(fli1:EGFP)$^{y1}$* embryos at 3 dpf (Fig 3A and B), recapitulating the *vegfc* MO induced phenotype (Fig 3A and C) and the previously

reported phenotype of *vegfc* mutants (Villefranc et al, 2013). In addition, *bach2a* morphants failed to express the lymphatic endothelial marker *lyve1*, but the expression of *vegfc* receptor *flt4* remained unchanged (Fig S5). To rule out the possibility that the observed phenotypes are a consequence of developmental delay, we assessed the formation of the TD at 4 dpf. MO-mediated down-regulation of *bach2a* resulted in a significant decrease in TD formation (76%, Fig 4A and B) as compared with control MO–injected siblings. Similarly, 87% of *vegfc* morphants were devoid of a TD (Fig 4A and C), as previously reported (Villefranc et al, 2013; Shin et al, 2016). In contrast to *bach2a* morphants, *bach2b* MO (Figs 3A and D and 4A and D) and *bach2b* gRNA-injected *Tg(fli1:EGFP)$^{y1}$* embryos (Figs 3A and E and 4A and E) exhibited no lymphatic defects. In line with the

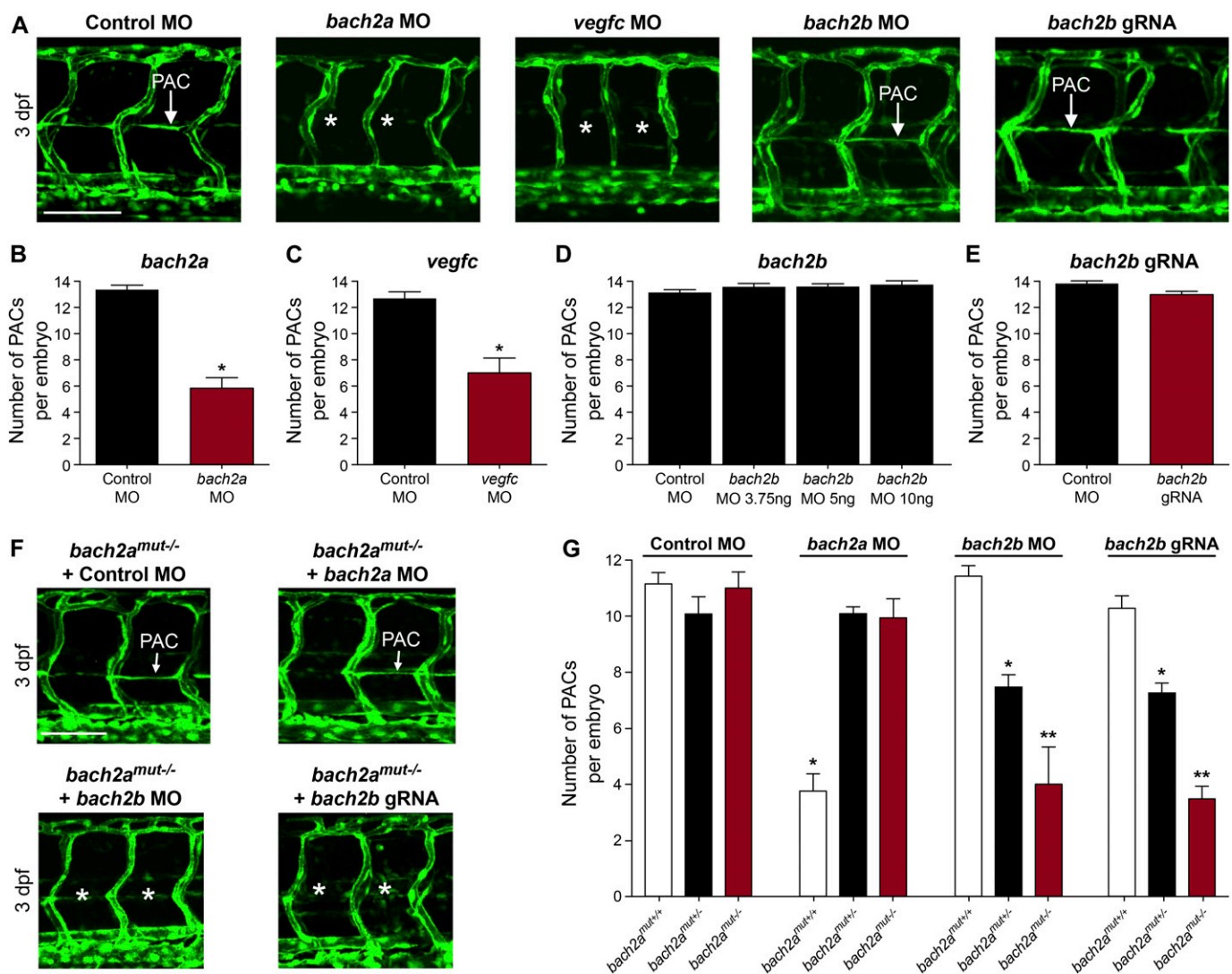

**Figure 3.** ***bach2a*** **is essential for parachordal cell (PAC) development in zebrafish embryos.**

**(A, B)** Confocal projection of the trunk of 3-dpf *Tg(fli1:EGFP)^{y1}* embryos showing PACs (white arrow) in control MO, *bach2b*, or *bach2b* gRNA-injected embryos but not after injection with *bach2a or vegfc* MO (white asterisk). Scale bar, 100 μm. (B) Number of PAC-containing segments (mean ± SEM) in 3-dpf *Tg(fli1:EGFP)^{y1}* zebrafish embryos injected with control MO (10 ng) or *bach2a* MO (3.75 ng, $n_{Control MO}$ = 46; *bach2a* MO, $n_{bach2a MO}$ = 53; \*$P$ < 0.001). Error bars, mean ± SEM. **(C)** Number of PAC-containing segments in *vegfc* MO-injected morphants (10 ng, $n_{control MO}$ = 53, $n_{vegfc MO}$ = 41; \*$P$ < 0.0001). Error bars, mean ± SEM. **(D)** Number of PAC-containing segments in 3-dpf *Tg(fli1:EGFP)^{y1}* zebrafish embryos injected with indicated *bach2b* MO concentrations (3.75, 5, or 10 ng, $n_{Control MO}$ = 18; $n_{bach2b MO-3.75ng}$ = 30; $n_{bach2b MO-5ng}$ = 30; $n_{bach2b MO-10ng}$ = 10; $P$ ≥ 0.2819). Error bars, mean ± SEM. **(E)** Quantification of PAC-containing segments in embryos injected with *bach2b* gRNA (125 ng, $n_{Control MO}$ = 35 $n_{bach2b gRNA}$ = 42; $P$ = 0.0615). Error bars, mean ± SEM. **(F)** Confocal projection of the trunk region showing PAC-containing segments in *Tg(fli1:EGFP)^{y1}*-homozygous *bach2a* mutants (*bach2a^{mut-/-}*) from F2 *bach2a^{mut+/-}* incross. An asterisk indicates the absence of PACs in *bach2a^{mut-/-}* embryos injected with *bach2b* MO (*bach2a^{mut-/-}* + *bach2b* MO) or *bach2b* gRNA (*bach2a^{mut-/-}* + *bach2b* gRNA) and a white arrow, their presence. Scale bar, 100 μm. **(G)** Number of PAC-containing segments (mean ± SEM) in 3-dpf embryos randomly selected from *bach2a^{+/-}* F2 incross progeny injected with control MO (10 ng, $n_{bach2amut + Control MO}$ = 50; $P$ = 0.5514), *bach2a* MO (3.75 ng, $n_{bach2amut + bach2a MO}$ = 75; \*$P$ < 0.0001), *bach2b* MO (3.75 ng, $n_{bach2amut + bach2b MO}$ = 55; \*$P$ and \*\*$P$ < 0.0222), or *bach2b* gRNA (125 ng, $n_{bach2amut + bach2b gRNA}$ = 137; \*$P$ and \*\*$P$ < 0.0001). After genotyping, offspring followed the expected Mendelian ratios of inheritance. **(B, C, D, E, G)** Wilcoxon rank sum test in panels (B, C, E) and Kruskal–Wallis test in panels (D, G).

possible genetic compensation underlying the absence of a clear angiogenic phenotype at early developmental stages (Fig 2D and E), injection of either a sub-dose of *bach2b* MO or *bach2b* gRNA into the progeny of *bach2a^{+/-}* intercross leads 23% of embryos to display significant lymphangiogenic defects, which were subsequently identified by genotyping as homozygous mutants (*bach2a^{mut-/-}* + *bach2b* MO or *bach2a^{mut-/-}* + *bach2b* gRNA) (Figs 3F and G and 4F and G). In contrast, control MO–injected *bach2a^{mut-/-}* appeared normal (Figs 3F and G and 4F and G). Likewise, no lymphatic phenotype was

detected in *bach2a^{mut-/-}* embryos upon injection of *bach2a* MO (Figs 3F and G and 4F and G), indicating that *bach2a* MO has minimal off-target effects. Finally, cardiac and body edema, characteristic of lymphatic-related defects, were detected in *bach2a* mutants after *bach2b* MO or *bach2b* gRNA injection, but not upon *bach2a* MO administration (Fig S6). Remarkably, the phenotypic defects resulting from *bach2a* down-regulation were more pronounced than those caused by the loss of *vegfc*, possibly due to the involvement of *bach2a* in additional signaling pathways (Zhou et al, 2016).

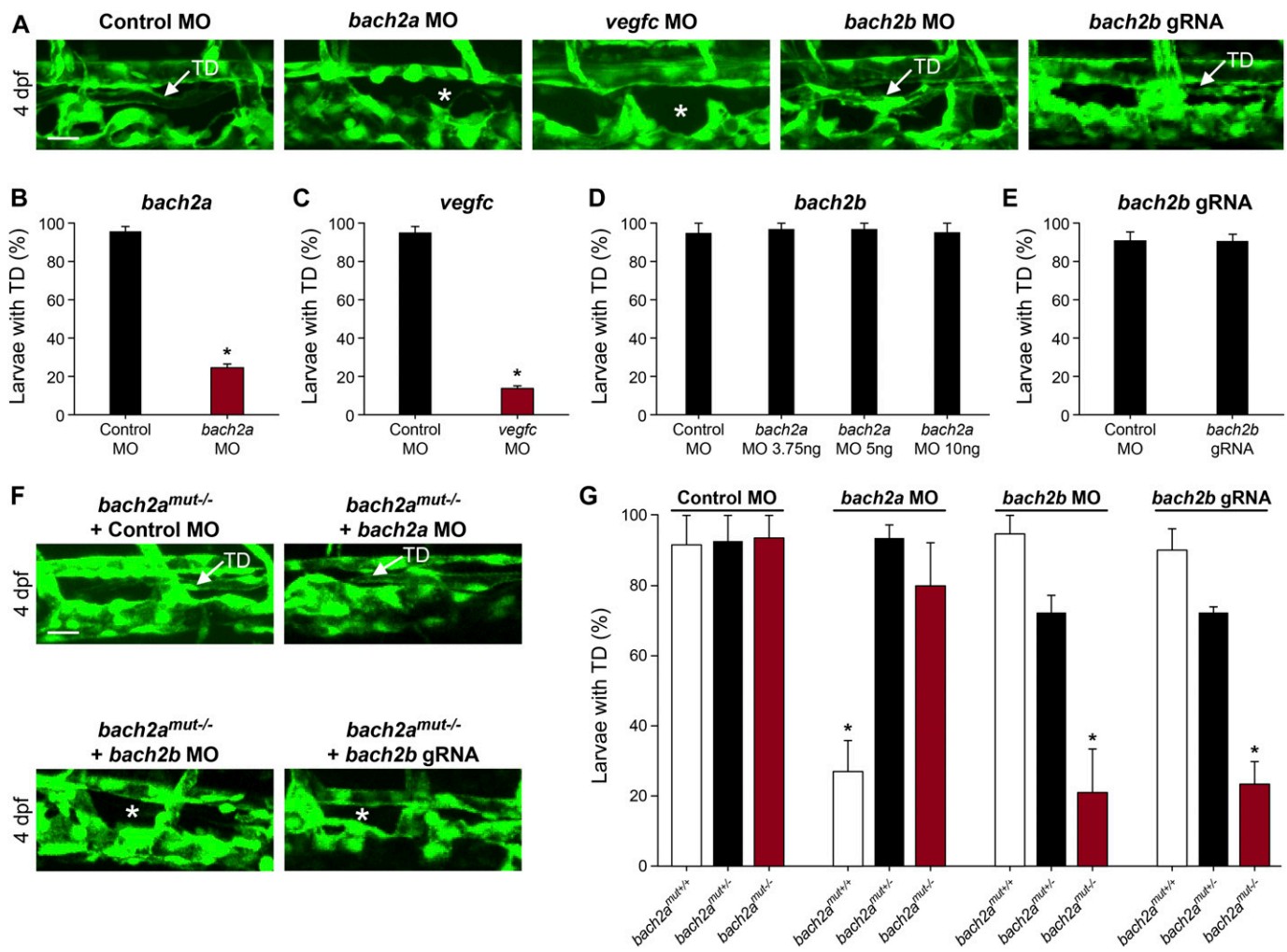

**Figure 4. *bach2a* is necessary for thoracic duct (TD) development in zebrafish larvae.**
**(A)** Confocal images of the TD in 4-dpf control MO- (10 ng), *bach2a* MO-(3.75 ng), *vegfc* MO- (10 ng), *bach2b* MO- (10 ng), or *bach2a* gRNA-injected *Tg(fli1:EGFP)^y1* larvae. A white arrow indicates the presence of a TD and a white asterisk, its absence. Scale bar, 20 µm. **(B)** Percentage of 4-dpf *Tg(fli1:EGFP)^y1* larvae with an intact TD after injection with control MO (10 ng) or *bach2a* MO (3.75 ng, $n_{Control\ MO}$ = 46; $n_{bach2a\ MO}$ = 52). Error bars, mean ± SEM; *$P$ < 0.0001. **(C)** Percentage of larvae with a TD after injection with *vegfc* MO (10 ng, $n_{Control\ MO}$ = 46; $n_{vegfc\ MO}$ = 52). Error bars, mean ± SEM; *$P$ < 0.0001. **(D)** Quantification of *bach2b* morphants with an intact TD after injection with indicated *bach2b* MO concentrations (3.75, 5, or 10 ng, $n_{Control\ MO}$ = 18; $n_{bach2b\ MO-3.75ng}$ = 30; $n_{bach2b\ MO-5ng}$ = 30; and $n_{bach2b\ MO-10ng}$ = 10; $P$ ≥ 0.1356). Error bars, mean ± SEM. **(E)** Percentage of TD-containing larvae injected with *bach2b* gRNA (125 ng, $n_{Control\ MO}$ = 35 $n_{bach2b\ gRNA}$ = 42; $P$ = 0.5461). Error bars, mean ± SEM. **(F)** Confocal images of a TD (white arrow) in 4-dpf *Tg(fli1:EGFP)^y1*: homozygous *bach2a* mutants (*bach2a*^mut−/−) derived from bach2a^mut+/− F2 incross. An asterisk indicates absence of a TD and a white arrow, its presence. Scale bar, 20 µm. **(G)** Analysis of *Tg(fli1:EGFP)^y1* 4-dpf progeny obtained from F2 *bach2a* heterozygous intercross. Random selection from the pool of siblings injected at the one-cell stage with control MO (10 ng, $n_{bachmut + Control\ MO}$ = 50), *bach2a* MO (3.75 ng, $n_{bach2amut + bach2a\ MO}$ = 75), *bach2b* MO (3.75 ng, $n_{bach2amut + bach2b\ MO}$ = 55), or *bach2b* gRNA (125 ng, $n_{bach2amut + bach2b\ gRNA}$ = 137) was found to maintain, after genotyping, the expected Mendelian ratios of inheritance. Error bars, mean ± SEM; *$P$ < 0.0003. **(B, C, D, E, G)** Wilcoxon rank sum test in panels (B, C, E) and Kruskal–Wallis test in panels (D, G).

Taken together, these findings demonstrate that knockdown of *bach2a* through MO injection results in the formation of defective blood and lymphatic vascular plexuses, despite the presence of the *bach2b* paralog, indicating that *bach2a* is a major contributor to this phenotype. However, knockout of *bach2a* expression (in mutants) triggers a functional compensation by *bach2b*, suggesting that the two *bach2* genes may share partially overlapping functions, which allows them to compensate for each other's loss during blood and lymph vessel development. Accordingly, it was suggested that BACH1 and BACH2 act in a complementary manner to maintain normal alveolar macrophage function and surfactant homeostasis in the lung (Ebina-Shibuya et al, 2016). Similar phenotypic differences between mutants and transient knockdown animals were observed in various model systems, some of which were attributed to genetic compensation (Rossi et al, 2015; El-Brolosy & Stainier, 2017; El-Brolosy et al, 2019).

## BACH1 promotes angiogenesis and lymphangiogenesis during tumor expansion in mouse models

In light of the perception that cancers frequently reactivate embryonic developmental signaling cascades to promote their expansion and aggressiveness, resulting in metastasis and poor patient outcome, we decided to evaluate the contribution of BACH to vascular remodeling during tumor progression. Emerging evidence point to BACH1, a

ubiquitously expressed protein, as a tumor-promoting factor that acts via multiple intracellular signaling cascades (Alvarez & Woolf, 2011; Yun et al, 2011). Recently, higher levels of BACH1 were found to be associated with poor prognosis in human ovarian cancer (Han et al, 2019) and to promote lung cancer metastasis (Lignitto et al, 2019; Wiel et al, 2019). BACH1 has been established as a major regulator of breast cancer bone metastasis (Liang et al, 2012) and was postulated as a potential novel therapeutic candidate for cancer treatment (Davudian et al, 2016a; Lee et al, 2019). To assess whether enhanced expression of BACH1 in tumor cells can stimulate blood and lymphatic vessel expansion during tumor progression as well as metastatic spread, we analyzed various ovarian and lung mouse tumors (Figs 5 and S7). Inoculation of human ovarian clear cell carcinoma ES2 cells into the

peritoneal cavity of immune-deficient CD-1 nude female mice has been established as a model of metastatic ovarian cancer (Shaw et al, 2004). In this study, we used this model to evaluate the effects of ectopic expression of BACH1 in human ovarian ES2 carcinoma cells (Fig 5). 20 d post-injection, metastases were detected in the diaphragm of animals injected with BACH1-overexpressing ES2 cells. An increase in CD34+ tumor-associated blood vessel density was apparent in diaphragm specimens derived from mice injected with BACH1-overexpressing cells, as compared with control ES2 cells (Fig 5A and B). Similar differences in the intratumoral blood vasculature were observed after subcutaneous inoculation of either BACH1-overexpressing ES2 cells (Fig S7A–C) or Bach1-overexpressing mouse D122 Lewis Lung Carcinoma cells (Eisenbach et al, 1984) (Fig S7E–G). These

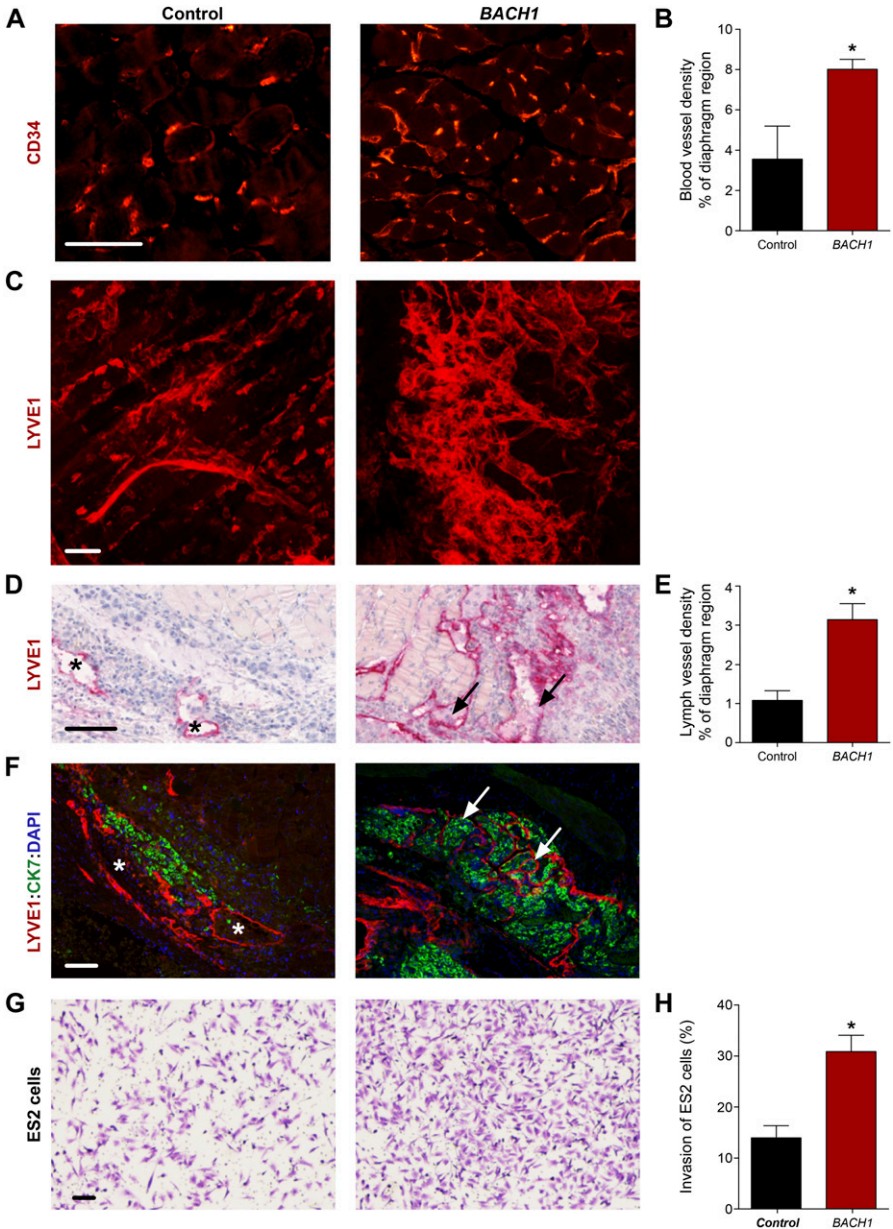

**Figure 5. *BACH1* promotes angiogenesis and lymphangiogenesis during ovarian tumor progression in mouse models.**
Ex vivo analysis of subcutaneous xenografts and diaphragm specimens excised from CD-1 nude female mice implanted with control (Control) or *BACH1* ectopically expressing (*BACH1*) human ovarian clear cell carcinoma ES2 cells. **(A)** Immunofluorescence labeling of blood vessels using anti-CD34 antibodies in diaphragm specimens excised from mice injected intraperitoneally with control or *BACH1*-expressing human ES2 cells. Scale bar, 100 $\mu$m. **(B)** Morphometric analysis of the diaphragm relative region covered by CD34+ blood vessels. Diaphragms were excised from mice inoculated intraperitoneally with either control (Control, n = 3) or *BACH1*-overexpressing (*BACH1*, n = 7) ES2 cells (mean ± SEM; *P = 0.0304). **(C)** Confocal z-projection images (Z dimension 7 $\mu$m) of control and *BACH1*-overexpressing subcutaneous-ES2 ovarian carcinoma xenografts subjected to LYVE1 immunofluorescence staining along with a modified CLARITY technique. Images demonstrate the complexity of the lymphatic vasculature. Scale bar, 100 $\mu$m. 3D reconstructions of the stacks are available in Videos 1 and 2. **(D)** Lymphatic vessel immunostaining, using anti-LYVE1 antibodies, of diaphragm specimens excised from mice injected intraperitoneally with control or *BACH1*-overexpressing ES2 ovarian carcinoma cells. A black arrow indicates infiltration of cells into the lymphatic vessel and an asterisk, their absence. Scale bar, 100 $\mu$m. **(E)** Morphometric analysis of the diaphragm relative region covered by LYVE1+ lymph vessels. Diaphragms were excised from mice inoculated intraperitoneally with either control (Control, n = 6) or *BACH1*-overexpressing (*BACH1*, n = 9) ES2 cells (mean ± SEM; *P = 0.0047). **(F)** Immunofluorescence double staining of LYVE1+ lymphatic vessels (red) and cytokeratin 7 (CK7, green) of a 4-$\mu$m-thick specimen sectioned from paraffin-embedded diaphragm excised from mouse inoculated intraperitoneally with either control or *BACH1*-overexpressing ES2 cells. Nuclei were counterstained with DAPI (blue). A white arrow indicates infiltration of tumor cells into the lymphatic vessels and an asterisk, their absence. Scale bar, 100 $\mu$m. **(G)** Transwell Matrigel invasion assay performed in vitro with ES2 ectopically expressing *BACH1* and control cells. The crystal violet dye staining images of the lower chambers are shown. Scale bar, 100 $\mu$m. **(G, H)** Percentage of ES2 cells that invaded through the Matrigel matrix (as in panel G) normalized to total cell number (n = 2 for each group, in duplicates; mean ± SEM; *P = 0.0209). **(B, E, H)** Wilcoxon rank sum test in panels (B, E, H).

results indicate that, in addition to its role in embryonic vascular development, BACH1 promotes tumor angiogenesis.

Numerous reports suggest that, in addition to blood vessel formation, lymphangiogenesis and lymphatic vessel remodeling are pivotal events for tumor expansion and metastatic spread (Stacker et al, 2014). We, thus, examined the effects of BACH1 on the lymphatic vasculature in various mouse tumor models. To gain insight into the 3D complexity of the lymphatic network within intact tumors, we subjected ES2 subcutaneous xenografts to a modified CLARITY (Clear, Lipid-exchanged, Anatomically Rigid, Imaging-compatible, Tissue hYdrogel) technique (Hama et al, 2011; Chung et al, 2013; Oren et al, 2018), along with LYVE1 immunofluorescence staining. A robust increase in lymphatic vessel density was observed in BACH1-overexpressing tumors as compared with control-derived ES2 tumors (Fig 5C and Videos 1 and 2). Morphometric analyses indicate a significant increase in the relative area fraction occupied by LYVE1⁺ tumor-associated lymphatic vessels in diaphragm specimens removed from mice injected with BACH1-overexpressing ES2 cells (Fig 5D and E). In addition, enlarged lymph vessels were observed in the peritumoral region of subcutaneous tumors derived from BACH1-overexpressing ES2 and D122 cells (Fig S7D and H, respectively). Expansion of the lymph vessels in the diaphragms of BACH1-overexpressing tumors

was associated with increased cell infiltration (arrow, Fig 5D). Co-immunofluorescence using antibodies directed against LYVE1 and cytokeratin 7, an antigen expressed in ES2 cells (Stimpfl et al, 1999), revealed that theᵽ metastasizing cells originated from the transplanted ES2 tumor cells (arrow, Fig 5F). This potential of BACH1 to promote tumor cell invasion was further confirmed by an in vitro invasion assay demonstrating an approximately twofold increase in the invasion abilities of BACH1-overexpressing as compared with control ES2 cells (Fig 5G and H). These data indicate that in mouse models, BACH1 remodels vascular architecture and promotes metastatic spread of tumor cells via the lymphatic vessels.

## BACH and VEGFC are functionally linked

Next, we investigated the molecular mechanisms underlying BACH activity. Because BACH proteins can directly bind to the promoter of their target genes (Ogawa et al, 2001; Warnatz et al, 2011; Yun et al, 2011), we assessed the ability of the binding sites in the VEGFC promoter region identified in silico (Figs 1A and 6A) to interact with BACH. Chromatin immunoprecipitation (ChIP) was performed on human ovarian clear cell carcinoma ES2 cells, overexpressing BACH1 tagged with a human influenza HA tag (Fig S7A). We observed that specific

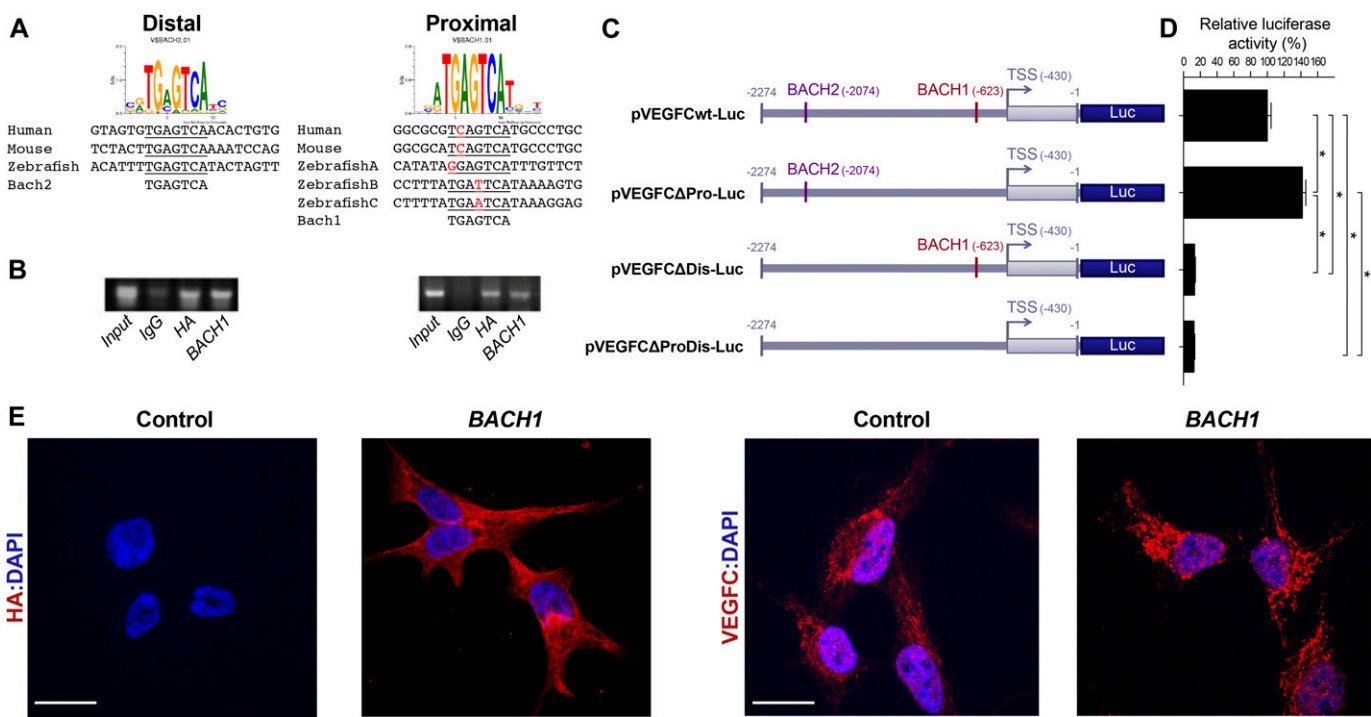

**Figure 6. BACH1 and *VEGFC* genetically interact.**
**(A)** Conservation of BACH sites in human, mouse, and zebrafish. The distal site is completely conserved. The proximal site is fully conserved between mouse and human, whereas there are three BACH sites at very close proximity in zebrafish. All proximal sites differ by one nucleotide from the consensus sequence. **(B)** Chromatin immunoprecipitation assay, followed by PCR measurements, was performed using primer mapping to the above human BACH proximal and distal regulatory sites and DNA precipitated with nonspecific IgG, HA-tag, or BACH1 antibodies. **(C)** Schematic representation of the wild-type human VEGFC promoter-driven luciferase (Luc) reporters (blue) (pVEGFCwt-Luc) and of three constructs deleted either from proximal (nt. −623 to −603, pVEGFCΔPro-Luc) or distal (nt. −2074 to −2054, pVEGFCΔDis-Luc) BACH-binding sites or a combination thereof (nt −623 to −603 and −2074 to −2054, pVEGFCΔProDis-Luc). Numbers refer to the nucleotide positions relative to ATG (translation initiation). **(D)** Quantification of dual–luciferase activity in human ES2 cells driven from pVEGFCwt-Luc, pVEGFCΔPro-Luc, pVEGFCΔDis-Luc, and pVEGFCΔProDis-Luc constructs. Relative luciferase activity is shown as a percentage of the pVEGFCwt-Luc value (mean ± SEM, *n* = 3). *P < 0.0001, Kruskal–Wallis test. **(E)** Immunofluorescence staining of human ES2 cells stably expressing either an empty pIRES vector (Control) or N-terminally HA-tagged *BACH*1 (*BACH1*) with antibodies directed against the HA tag (red, left panel) or against VEGFC (red, right panel). Nuclei were counterstained with DAPI (blue). Scale bar, 20 *μm*. TSS, transcription start site.

sequences from both proximal and distal regulatory regions were enriched in the presence of antibodies against either HA or BACH1, indicating that both sites are transcriptionally functional (Fig 6B). To investigate the functional significance of two potential BACH-binding sites, promoter–reporter constructs were engineered with deletion of the BACH-binding sites, either separately or concurrently. Disruption of the proximal site led to a 23% increase in *VEGFC* basal promoter activity (Fig 6C and D), supporting an inhibitory role. In contrast, deletion of the distal binding site individually or in combination with the proximal site resulted in an 86% reduction in *VEGFC* basal promoter activity (Fig 6C and D), suggesting a pivotal role in basal promoter functioning. In addition, overexpression of *BACH1* in human ovarian ES2 carcinoma cells induces a significant increase in VEGFC

expression (Fig 6E), strongly supporting a genetic and functional link between these two factors.

To investigate whether these results are recapitulated in vivo, we analyzed the effect of *bach2a* knockdown on *vegfc* expression during zebrafish development. A marked down-regulation of *vegfc* expression was observed in *bach2a* morphants (Fig 7A). This effect was specific for *bach2a*, as *vegfc* expression remained intact in *bach2b* MO-injected embryos (Fig S8). We then attempted to rescue the phenotypes of *bach2a* morphants by overexpressing *vegfc* mRNA. Co-injection of in vitro–transcribed *vegfc* mRNA and *bach2a* MO partially restored PHBC formation at 30 hpf (Fig 7B and C). Similarly, the number of PAC-containing segments increased by ~50% after co-injection of *bach2a* MO and *vegfc* mRNA (Fig 7D and E) and a twofold recovery in

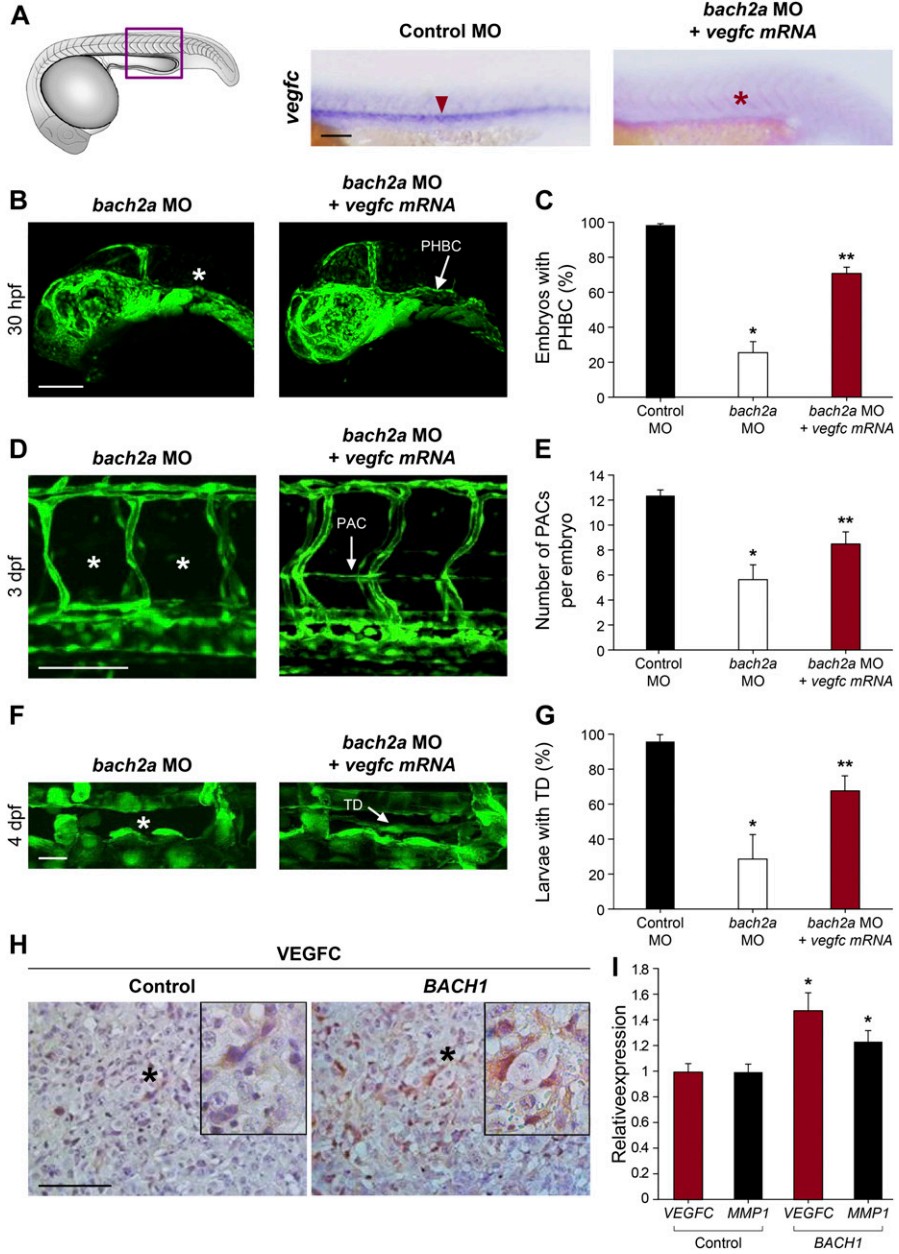

Figure 7. **BACH mediates angiogenesis and lymphangiogenesis in a VEGFC-dependent manner.**
**(A)** Whole-mount in situ hybridization of 24-hpf, wild-type zebrafish embryos demonstrating the expression of *vegfc* mRNA after injection with control MO (Control MO 10 ng, red arrowhead) and the absence of its expression in embryos injected with specific MO targeting *bach2a* (3.75 ng, *bach2a* MO, red asterisk). Scale bar, 100 μm. **(B)** Confocal images of 30-hpf *Tg(fli1:EGFP)$^{y1}$* embryos co-injected with specific MOs targeting *bach2a* (3.75 ng) and in vitro–transcribed *vegfc* mRNA (800 pg, *bach2a* MO + *vegfc* mRNA) demonstrating the restoration of PHBC (white arrow). **(C)** Percentage of rescued PHBC defects in 30-hpf *Tg(fli1:EGFP)$^{y1}$* embryos after co-injection with *bach2a* MO (3.75 ng) and *vegfc* mRNA (800 pg, $n_{Control\ MO}$ = 56; $n_{bach2a\ MO}$ = 58; $n_{bach2a\ MO\ +\ vegfc\ mRNA}$ = 56). Error bars, mean ± SEM; * or **$P$ < 0.0001. **(D)** Rescue of parachordal cell (PAC) development in 3-dpf *Tg(fli1:EGFP)$^{y1}$* embryos after the co-injection of *bach2a* MO and *vegfc* mRNA (*bach2a* MO + *vegfc* mRNA; PACs are indicated by a white arrow). Scale bar, 100 μm. **(E)** Quantification of the number of PAC-containing segments (mean ± SEM) in 3-dpf *Tg(fli1:EGFP)$^{y1}$* embryos after the co-injection of *bach2a* MO (3.75 ng) and *vegfc* mRNA ($n_{Control\ MO}$ = 47; $n_{bach2a\ MO}$ = 27; $n_{bach2a\ MO\ +\ vegfc\ mRNA}$ = 54; * or **$P$ < 0.01). **(F)** Thoracic duct (TD) formation in 4-dpf *Tg(fli1:EGFP)$^{y1}$* embryos co-injected with *bach2a* MO and *vegfc* mRNA (*bach2a* MO + *vegfc* mRNA; TD is indicated by a white arrow). Scale bar, 20 μm. **(G)** Percentage of 4-dpf *Tg(fli1:EGFP)$^{y1}$* embryos showing normal TD after *bach2a* MO and *vegfc* mRNA injection (*bach2a* MO + *vegfc* mRNA). ($n_{Control\ MO-10ng}$ = 32; $n_{bach2a\ MO}$ = 65; $n_{bach2a\ MO\ +\ vegfc\ mRNA}$ = 68 (Error bars, mean ± SEM; * or **$P$ < 0.01. **(H)** Immunohistochemistry labeling of control (Control) or *BACH1* (*BACH1*) ectopically expressing ES2 ovarian carcinoma xenograft specimens using anti-VEGFC antibodies and counterstaining with hematoxylin (blue). Black asterisk localizes the region magnified in the black frame. Scale bar, 100 μm. **(I)** Quantitative RT-PCR measurement of *VEGFC* and *MMP1* mRNA expression in xenograft initiated either from control or *BACH1*-overexpressing ES2 cells (n = 5 in each group; mean ± SEM; *$P$ < 0.05. **(C, E, G, I)** Kruskal–Wallis test in panels (C, E, G) and Wilcoxon rank sum test in panel (I).

TD formation was detected (Fig 7F and G). In addition, pericardial and body edema, as well as reduced blood flow, were partially restored after an injection of *vegfc* mRNA (Fig S9).

Finally, we addressed the molecular basis of the BACH1 and VEGFC interaction during tumor progression in the various mouse tumor models. VEGFC expression was specifically elevated in BACH1-overexpressing ES2 and D122 tumors both at the protein (Figs 7H and S7J, respectively) and mRNA level (Figs 7I and S7I), with no significant changes in the mRNA levels of either VEGFA or VEGFB (Fig S10). Similarly, the mRNA expression levels of *MMP1*, a recognized transcription target of BACH1 (Yun et al, 2011; Liang et al, 2012), were significantly elevated in subcutaneous *BACH1*-overexpressing ES2 xenografts (Fig 7I). Collectively, these results highlight BACH as a novel regulator of blood and lymphatic vessel formation during both embryonic development and mouse tumor expansion, placing it upstream of VEGFC in these cascades.

### Expression of *BACH1* and *VEGFC* correlates during human cancer progression

Metastatic spread of cancer cells from primary solid tumors to sentinel lymph nodes and distant tissues and organs is one of the hallmarks of malignant neoplasms (Valastyan & Weinberg, 2011), responsible for most human cancer-related deaths. Neoplastic cell dissemination may occur either via blood vessels or via the lymphatic system (Paduch, 2016). VEGFC is one of the key factors promoting malignant cell spread, as demonstrated both in mouse tumor models (Mandriota et al, 2001; Skobe et al, 2001; Ma et al, 2018) and during human cancer progression (Thiele & Sleeman, 2006; Rinderknecht & Detmar, 2008; Chen et al, 2012; Jiang et al, 2014). Similarly, it was shown that BACH1 promotes the metastasis of breast cancer through different molecular mechanisms (Liang et al, 2012; Lee et al, 2013; Lee et al, 2014) and its stabilization in lung adenocarcinoma is associated with increased metastatic dissemination and poor survival (Lignitto et al, 2019; Wiel et al, 2019). To substantiate the pathophysiological relevance of the interaction between *BACH1* and *VEGFC*, we carried out an in silico analysis of publicly available gene-expression data generated by The Cancer Genome Atlas Research Network (https://www.cancer.gov/about-nci/organization/ccg/research/structural-genomics/tcga). We analyzed RNA sequencing–derived data regarding aberrant gene expression in specimens taken from melanoma and lung adenocarcinoma (LUAD) human cancer patients. Melanoma is one of the commonest forms of skin cancer, whereas LUAD is at present the most common lung cancer subtype among nonsmokers. In both cancers, although the early-stage is curable by surgical resection, lymphatic metastasis results in poor prognosis. Interestingly, we found that *BACH1* expression positively and significantly correlates with the expression of *VEGFC* in human melanoma and LUAD cancer progression (Fig 8). Specifically, *BACH1* and *VEGFC* expression are significantly higher in samples from melanoma patients clinically diagnosed with lymph node metastatic spread, as compared with those with primary tumors (Fig 8A). Similarly, the expression of *BACH1* and *VEGFC* were significantly augmented in specimens derived from clinical stage III LUAD patients in comparison with stage I and II (Fig 8B). According to the tumor node metastasis taxonomy classification, stage I refers to the early, nonmetastatic stage, whereas stages II and III usually indicate the intermediate, regional lymphatic metastatic stages, of which stage III has a higher lymphatic metastasis degree than stage II. This

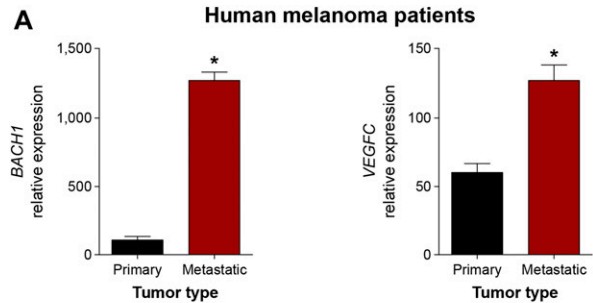

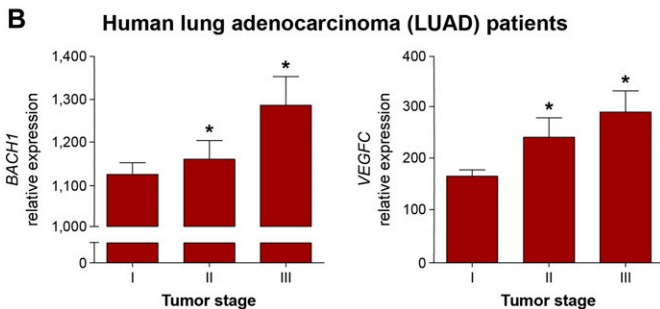

**Figure 8. Expression of *BACH1* and *VEGFC* correlates during human cancer progression.**

**(A)** Correlation of *BACH1* and *VEGFC* expression in specimens from melanoma patients clinically diagnosed with a primary tumor (n = 84) in comparison with those with metastatic stage melanoma (n = 356), as deduced from The Cancer Genome Atlas Research Network RNA sequencing (RNA-Seq) data. Mean ± SEM; *P ≤ 0.0008. **(B)** Correlation of *BACH1* and *VEGFC* expression in specimens from lung adenocarcinoma (LUAD) clinically diagnosed with primary tumor stages I (n = 292), II (n = 133), and III (n = 95), as analyzed from The Cancer Genome Atlas Research Network RNA-Seq. data. Mean ± SEM; *P < 0.02. **(A, B)** Wilcoxon rank sum test in panel (A) and Kruskal–Wallis test in panel (B).

correlation may indicate the potential ability of *BACH1* and *VEGFC* to promote cellular migration and cancer invasion. Hence, BACH1 and VEGFC may serve as candidate diagnostic biomarkers in cancer patients.

## Discussion

VEGFC is a potent regulator of the growth and maintenance of blood and lymphatic vessels during embryonic development, tumor expansion, and metastasis. The results presented here demonstrate that members of the BACH family regulate VEGFC expression, thereby promoting angiogenesis and lymphangiogenesis during zebrafish development and in ovarian and lung mouse tumor models.

BACH1 and BACH2 are important for the homeostasis of heme, an essential molecule for many biological functions (Warnatz et al, 2011; Igarashi & Watanabe-Matsui, 2014). Imbalanced levels of heme can cause oxidative stress when it reacts with molecular oxygen, which in turn disrupts various cellular signaling pathways. BACH1 was initially discovered as a physiological repressor of heme oxygenase-1, a rate-limiting enzyme in heme catabolism stimulated by nuclear factor erythroid 2-related factor 2 (NRF2) (Warnatz et al, 2011). Recent emerging evidence, however, indicate the widespread functions of BACH1 in diverse physiological and pathological settings, including hematopoiesis, inflammation, cardiovascular disease, aging, metabolic disorders, and neurodegenerative diseases. BACH2 was established as a crucial factor

for B- and T-cell memory differentiation. Nevertheless, the potential roles of these transcription factors during embryogenesis remain obscure (Sidwell & Kallies, 2016). Members of the BACH family possess both a BTB/POZ and a CNC-bZip functional domain, conferring them with protein–protein interaction and DNA-binding capabilities (Zhou et al, 2016) to the antioxidant response elements (Zhou et al, 2016). The BTB/POZ domain, located at the N terminus, mediates dimer formation (Igarashi & Watanabe-Matsui, 2014). Conversely, the bZip domain enables DNA binding and the formation of heterodimers with small musculoaponeurotic fibrosarcoma (sMaf) transcription factors (Davudian et al, 2016a). Three evolutionary conserved sMafs have been identified in mammals and an additional one in zebrafish, all displaying complex expression patterns during embryogenesis (Yamazaki et al, 2012). Whereas single sMaf knockout mice show no or a mild phenotype, triple-knockout embryos display severe growth retardation and liver hypoplasia, resulting in embryonic lethality at E13.5 (Yamazaki et al, 2012). Interestingly, a member of the large Maf family of transcription factors, mafba in zebrafish (Koltowska et al, 2015) and Mafb in mice (Dieterich et al, 2015), was shown to be activated by VEGFC. In zebrafish, mafba is crucial for the migration of lymphatic precursors after their initial sprouting from the cardinal vein (Koltowska et al, 2015). In LECs, MAFB promotes the expression of PROX1, KLF4, NR2F2, and SOX18, key transcription factors and markers of differentiated LECs. Furthermore, E14.5 Mafb$^{-/-}$ mouse embryos show impaired lymphatic patterning in the skin (Dieterich et al, 2015). Altogether, these studies suggest that the BACH and MAF transcription factor families play key roles during lymphangiogenesis. Uncovering the complexity within these molecular networks may be exploited for the understanding of EC differentiation and vascular development.

Four bach genes (bach1a, bach1b, bach2a, and bach2b) with homology to mammalian BACH are present in the zebrafish genome. Phylogenetic analysis indicates that bach1a, bach1b and bach2a, bach2b belong to two distinct groups, bach1 and bach2, respectively, with bach2 possibly diverging earlier than bach1 (Luo et al, 2016). It was shown that, during embryogenesis, the two bach1 genes inhibit heme oxygenase 1a (hmox1a) induction in zebrafish (Fuse et al, 2015). In a recent study, overexpression of bach1b was found to suppress developmental angiogenesis by inhibiting Wnt/β-catenin signaling (Jiang et al, 2017). Herein, we show that the bach2a gene controls vegfc expression, directing blood and lymphatic vascular development in zebrafish. Yet, the lack of full overlapping expression between bach2a and vegfc supports additional tissue-specific functions for each of these factors.

Accumulating data establish BACH1 as a critical facilitator of tumorigenesis and metastasis in breast (Lee et al, 2013), colon (Davudian et al, 2016b), prostate (Shajari et al, 2018) ovarian (Han et al, 2019), and lung (Lignitto et al, 2019; Wiel et al, 2019) cancer. Elevated levels of BACH1 expression have been linked to a higher risk of breast cancer recurrence in patients (Liang et al, 2012), whereas association with metastatic spread and poorer prognosis has recently been suggested in the case of human ovarian cancer (Han et al, 2019) and lung adenocarcinoma (Lignitto et al, 2019; Wiel et al, 2019). Ectopic expression of BACH1 in breast cancer cells promotes malignancy and metastasis, whereas its knockdown suppresses these processes. BACH1 has been placed downstream of the Raf kinase inhibitory protein, a tumor suppressor gene shown to inhibit invasion and bone metastasis in a breast cancer xenograft mouse model (Yun et al, 2011; Lee et al, 2013). Inactivation of Raf kinase inhibitory protein during

tumor expansion results in higher expression of BACH1 and its target genes C-X-C chemokine receptor type 4 (CXCR-4) and matrix metalloproteinase1 (MMP1), established drivers of tumor progression and metastasis (Foley & Kuliopulos, 2014; Mishan et al, 2016). Furthermore, ablation of BACH1 in human colon carcinoma (Davudian et al, 2016b) or in prostate cancer cells (Shajari et al, 2018) prevents cell growth, migration, and invasion in vitro, decreasing the expression of its main metastasis-related genes, MMP1, let-7a, and CXCR4. Interestingly, cxcr4 is expressed in the somites and the endothelium of zebrafish embryos (Chong et al, 2001), where we detect the expression of both bach2a and bach2b transcripts. cxcr4 was shown to be crucial for lateral aortae formation (Siekmann et al, 2009) and trunk lymphatic vascular network assembly (Cha et al, 2012) and a key modulator of vascular progenitor cells (Sainz & Sata, 2007). In addition to its contribution to development, CXCR4 was shown to play a role in carcinogenesis, tissue repair and other pathological circumstances (Kawaguchi et al, 2019). VEGFC can, thus, be added to a growing list of tumor and metastatic proteins, including CXCR4 and MMP1, all of which are transcriptionally regulated by BACH1. Including VEGFC as a target of BACH1 allows novel perspectives of the role of BACH1 in vascular development during embryogenesis and pathological conditions.

During the past years, a substantial amount of in vitro, animal, and small-size human studies established BACH proteins as a hub of critical transcriptional networks that govern key processes during normal physiology and disease states. Surprisingly, however, Bach single-(Sun et al, 2002; Muto et al, 2004) or double- (Ebina-Shibuya et al, 2016) knockout mice exhibit no phenotype at birth. This may result from the existence of a yet unknown compensatory mechanism similar to the one described here in zebrafish. Further understanding the mechanisms underlying BACH's mode of action during embryogenesis, adult life, and tumorigenesis would pave the way to resolving this ambiguity.

Altogether, our results highlight BACH as a novel regulator of angiogenesis and lymphangiogenesis during embryonic development as well as tumor progression. We show that BACH controls the expression of VEGFC, an established pro-lymphangiogenic and angiogenic growth factor. The expanded recognition of BACH1 activity, together with its inhibitory effect on NRF2, an important detoxifying and antioxidant factor, marks it as a potential therapeutic target. Hence, various direct or indirect promising BACH1 modulators have been developed (Arbiser, 2011; Banerjee et al, 2013; Cuadrado et al, 2019). Accordingly, recent reports demonstrated that targeting of BACH1 and mitochondrial metabolism may serve as an effective therapy for triple negative breast cancer (Lee et al, 2019). In two new studies on lung cancer, it was shown that therapeutic intervention that either destabilizes BACH1 (Lignitto et al, 2019) or disrupts its ability to induce glycolysis (Wiel et al, 2019) have the potential to inhibit BACH1 pro-metastatic activity. Targeting the BACH1/VEGFC signaling axis with these inhibitors may potentially be significant therapeutically for various blood and lymphatic vessels pathologies as well.

# Materials and Methods

### Bioinformatics analysis

Transcription factor–binding site analysis was performed using the Genomatix Genome Analyzer (Genomatix Software GmbH) MatInspector

program (Cartharius et al, 2005). Promoter regions of the human, mouse, and zebrafish Vegfc genes were extracted from the University of California Santa Cruz genome browser (Kent et al, 2002) as follows: Human-GRC37/hg19 chr4: 177,713,306-177,716,211; Mouse-GRCm38/mm10 chr8:54,075,150-54,077,946; and Zebrafish Zv9/danRer7 chr1:39,270,126-39,273,225. The zebrafish *bach* sequences used are *bach1a*: NM_001040313.1; *bach1b*: NM_001020663.1; *bach2a*: XM_680223.9; and *bach2b*: XM_677841.6.

## Zebrafish lines and husbandry

The EK and *Tg(fli1:EGFP)$^{y1}$* (Lawson & Weinstein, 2002) zebrafish lines were maintained under standard conditions (Isogai et al, 2003). All experiments were carried out according to the guidelines of the Weizmann Institute Animal Care and Use Committee.

## FACS and RT-PCR analysis

*Tg(fli1:EGFP)$^{y1}$* embryos at 21–24 hpf and 3 dpf were collected and dissociated as previously described (Nicenboim et al, 2015). FACS of single-cell suspensions was performed at 4°C using FACS Aria flow cytometer (Becton Dickinson). Total RNA was isolated from equal numbers of GFP$^+$ and GFP$^-$ cells by PerfectPure RNA Cultured Cell kit (5 PRIME), according to the manufacturer's instructions. RNA (100 ng) from each sample was subjected to first-strand cDNA synthesis with Superscript III reverse transcriptase (Invitrogen) and random hexamers. All PCR conditions were optimized to produce a single product of the correct base pair size in the linear range of the reaction using the following set of primers: *bach1a*: 5′-TGTAAGACGGCGGAGTAAGA and 5′-CTTCAGCTGGTTGTGGTCT, *bach1b*: 5′-CTTCAGTGCTCGTGTGTCCA and 5′-TG-TAGGCGAACTCCAGCAAG, *bach2a*: 5′-GACAGAACACGAGCCACTCA and 5′-AC-AGCGCATGACATCTTGGA, *bach2b*: 5′-TGCATCCTGAACCTTGAGTGT and 5′-CT-GCACATCTCGACACACCT, *fli1*: 5′-CCGAGGTCCTGCTCTCACAT and 5′-GGGACT-GGTCAGCGTGAGAT, *bactin*: 5′-CGAGCAGGAGATGGGAACC and 5′-CAACGGA-AACGCTCATTGC.

## Construction of the pCS2*vegfc*CDS plasmid

The full-length coding region of zebrafish *vegfc* was amplified from cDNAs derived from 24 hpf embryos using a forward (5′-ATGCACT-TATTTGGATTTTCT) and a reverse (5′-TTAGTCCAGTCTTCCCCAG) primers, and sub-cloned into the pCRII-TOPO cloning vector (Invitrogen). After nucleotide sequence verification, a Gateway-compatible (Invitrogen) middle entry clone was generated using Gateway BP clonase (Invitrogen)–mediated recombination. A pCS2*vegfc*CDS plasmid was produced using Gateway LR clonase (Invitrogen). Capped mRNA was transcribed from a Not1-linearized template using the mMESSAGE mMACHINE Kit (Thermo Fisher Scientific).

## Microinjection of zebrafish embryos

Morpholino antisense oligonucleotides (Gene Tools) were designed with sequences complementary to zebrafish *bach2a* and *bach2b* cDNA in a location downstream to the initiating start codon. The morpholino sequences were as follows: *bach2a*: (GenBank accession number XM_680223.9), 5′-TGTCAGGCTTCTCCTCCATAGA-CAT-3′ and *bach2b*: (GenBank accession number XM_677841.6), 5′-CTTCAGACTTCTCATCCACGGACAT-3′. The morpholino targeting the zebrafish *vegfc* was previously described (Yaniv et al, 2006). As a control, the Gene Tools standard control MO (5′-CCTCTTACCTCAGTTA-CAATTTATA-3′) was applied. Specific MOs were injected into one-cell stage embryos at concentrations from 3.75 up to 10 ng and control MO at 10 ng per embryo. For rescue experiments, pCS2*vegfc*CDS mRNA (800 pg per embryo) was simultaneously injected with the morpholino (3.75 ng per embryo). For quantification of the phenotypes, embryos from each group were randomly selected. Quantification was at least three independent experiments.

## Design and synthesis of gRNA

The design of the *bach2a* and *bach2b* CRISPR guide was performed with CHOPCHOP (Montague et al, 2014) (chopchop.rc.fas.harvard.edu). Potential off-target sequences were checked using the MIT CRISPR Design site (crispr.mit.edu) (Hsu et al, 2013). Oligonucleotides synthesized for the guide sequence 5′-GGACGTCCTGTGTGACGTGA and 5′-CTGTGCCGAATTCCTGCGCA for *bach2a* and *bach2b*, respectively, were cloned into the BsmBI site of the pT7-gRNA plasmid (a gift from Wenbiao Chen) (plasmid # 46759; Addgene) (Jao et al, 2013). Alt-R S.p. Cas9 Nuclease 3NLS (250 ng/$\mu$l; Integrated DNA Technologies) and gRNA (125 ng/$\mu$l) in 5 $\mu$l total volume were co-injected into one-cell–stage *Tg(fli1:EGFP)$^{y1}$* embryos. To detect mutagenic events, DNA was extracted from 24-hpf embryos, amplified using a set of primers 5′-AGCAAGGAATGTCTATGGAGGA and 5′-ATGAGTGGCTCGTGTTCTGTC (235 bp) for *bach2a* and 5′-CGCTCCATTGTTACAGTTTGC and 5′-GCCGTCCTC-TTCACTGCGC (157 bp) for *bach2b*. PCR products were separated on 3.5% MetaPhor Agarose gel (Lonza). When both genotyping and phenotypic analyses of single zebrafish embryos from heterozygous *bach2a* intercross was needed, the larval tail (6 dpf) was used for genotyping, whereas RNA was extracted from the anterior part using NucleoSpin RNA Plus XS RNA purification kit (Macherey-Nagel).

## Whole-mount in situ hybridization

Embryos were dechorionated and fixed overnight in 4% PFA at 4°C, at the appropriate time points. In situ hybridization was performed as previously described (Yaniv et al, 2006) using single-stranded digoxygenin-dUTP–labeled RNA probes transcribed by T7 RNA polymerase (Roche). The PCR-generated probes were amplified with the following set of primers: *bach2a*: 5′-AAGAGTGAGCTAGAGGGCA and 5′-CGTTCTCTTGTTCGGGATCTTG; *bach2b*: 5′-CTGCGCAGTGAAGAGGACGG and 5′-GCTCCACCTCTTGCTTGCAC; *vegfc*: 5′-CATCAGCACTTCATACATCAGC and 5′-GTCCAGTCTTCCCCAGTATG; *lyve1*: 5′-GGTTTGGTTGGGTTGAGGAGC and 5′-TTAGGAAGAGTCAGAGTCTTGTTC; *flt4*: 5′-CTCGAGAATGACATGTGCTGG and 5′-CAGCCAGCGAGCACAAAGC. After performance of a color reaction with alkaline phosphatase substrates (Roche), embryos were fixed in 4% PFA and washed in PBS supplemented with 0.1% Tween-20 (PBST). For tissue clarification, embryos were mounted in glycerol and imaged using Leica M165 FC stereomicroscope.

## Cell culture

Human ovarian clear cell carcinoma ES2 cell line (American Type Culture Collection) was cultured in DMEM supplemented with 10% FBS. Mouse D122 Lewis lung carcinoma cell line (Eisenbach et al,

1984), kindly provided by Prof. Lea Eisenbach, (Weizmann Institute of Science, Israel), was grown in DMEM containing 10% FBS, 1 mM sodium pyruvate, and 1% nonessential amino acids. All cell lines were routinely tested for mycoplasma contamination using the EZ-PCR Mycoplasma Test Kit (Biological Industries).

## Establishment of stable BACH1-overexpressing cell pools

Full-length coding region of the human BACH1 (*BACH1*, GenBank accession number NM_206866.3) and mouse Bach1 (*Bach1*, GenBank accession number BC057894.1) with an *N*-terminal HA tag were reverse transcribed with SuperScript III Reverse Transcriptase (Invitrogen) and PCR-amplified using Phusion high-fidelity DNA polymerase (New England Biolabs) together with the following set of primers: *BACH1*: 5′-ATGTCTCTGAGTGAGAACTCGG and 5′-TTACTCATCAGTAGTACATTTATC; *Bach1*: 5′-ATGTCTGTGAGTGAGAGTGCG and 5′-TTACTCGTCAGTAGTGCACTTG. The fragments were ligated into pCRII-TOPO and their sequence fidelity was confirmed by sequencing. Inserts were restricted and ligated into pEIRES expression vector containing the human EF-1a promoter (Hobbs et al, 1998) to produce the pIRES*BACH1* and pIRES*Bach1* constructs. ES2 and D122 cells were transfected with pIRES*BACH1* and pIRES*Bach1* expression vectors, respectively, using Lipofectamine 2000 reagent (Invitrogen) and selected with 2.5 µg/ml puromycin (Sigma-Aldrich) as was previously described (Cohen et al, 2009). An average of 50 individual puromycin-resistant colonies were collected together, and BACH1 overexpression was confirmed by immunoblotting. ES2 and D122 cells that were stably transfected with the pEIRES empty vector were used as controls.

## ChIP assay

The ChIP experiments were performed using the ChIP-IT-Express kit (Active Motif Cat. no. 53009). Cells were fixed and cross-linked using 1% formaldehyde at room temperature for 10 min. Fixation was stopped by adding glycine. DNA extracted from nuclear fraction was subjected to enzymatic shearing for 35 min at 37°C to obtain mononucleosomes. The resulting chromatin preparation was immunoprecipitated with magnetic protein G–coupled beads and 10 mg of either anti HA-tag (HA.11; Covance), Bach1 (sc-14700; Santa Cruz), or nonspecific-IgG antibody. DNA–protein crosslinking was reversed at 65°C (4 h), treated with Proteinase K (2 h, 42°C). The recovered DNA was then subjected to PCR using the following set of specific primers: *BACH* proximal site: 5′-GAGGGAGAGTGAGAGGGG and 5′-CGCAGGATCCTCCAGAGC; *BACH* distal site: 5′-CCGAGTCTGA-TGGGATGGAA and 5′-GCCTTTGTTGATACAGCCTTGG.

## Transient transfection and luciferase assay

The 5′ regulatory region of human *VEGFC* gene (NG_034216.1) encompassing 2,274 nucleotides (−1 to −2,274 in relation to ATG, Fig 1) was synthesized by GenScript. This fragment was then used as a template for deletion of either the proximal (nt −623 to −603, 5′-GGCGCGTCAGT-CATGCCCTGC) or distal (nt −2074 to −2054, 5′-GTAGTGTGAGTCAA CACTGTG) BACH-binding site individually or in combination. Subsequently, these synthesized fragments were sub-cloned into the pGL4.10[*luc2*] promoter-less vector (Promega) between KpnI and XhoI restriction sites to generate the pVEGFCwt-Luc (wild-type VEGFC promoter region), pVEGFCΔPro-Luc (VEGFC promoter region deleted of the proximal BACH-binding site),

pVEGFCΔDis-Luc (VEGFC promoter region deleted of the distal BACH-binding site), and pVEGFCΔProDis-Luc (VEGFC promoter region deleted of both proximal and distal BACH-binding sites). All constructs were verified by DNA sequence analysis. For the luciferase reporter gene assay, ES2 cells plated in 24-well plates (40,000 cells per well) were co-transfected with the indicated pVEGFC-Luc construct in combination with the pRL-TK-Renilla (pRL-TK vector; Promega) luciferase internal control vector using Lipofectamine 2000 (Thermo Fisher Scientific), according to the manufacturer's instructions. After 48 h, the cells were lysed and luciferase activity was measured using the Dual-Luciferase Assay System (Promega). All luciferase data were corrected for transfection efficiency based on the Renilla internal control following by subtraction of the pGL4.10[*luc2*] activity background. The data were calculated as means ± standard error of three independent experiments, each performed in nine replications. Luciferase activities for each transfection are plotted as average fold-change in relation to the pVEGFCwt-Luc.

## Western blot analysis

Protein extraction was performed in RIPA buffer (20 mM Tris, pH 7.4, 137 mM NaCl, 10% glycerol, 0.5% [wt/vol] sodium deoxycholate, 0.1% [wt/vol] SDS, 1% Triton X-100, and 2 mM EDTA) supplemented with 1 mM PMSF and protease inhibitor cocktail (Sigma-Aldrich). Lysates (20 µg/lane) were electrophoresed in SDS–PAGE under reducing conditions and transferred to a nitrocellulose (Whatman). Membranes were probed with anti HA-tag monoclonal antibody (HA.11, 1:100; Covance), whereas β-tubulin (H-235, 1:500; Santa Cruz) was used as a loading control. Appropriate HRP-conjugated antimouse or antirabbit secondary antibodies (1:10,000; Jackson ImmunoResearch) were used, respectively.

## Cell invasion assay

The invasion potential of the cells was examined in vitro using BioCoat Matrigel Invasion Chamber (Corning). A total of $80 \times 10^3$ cells suspended in serum-free medium were seeded in the upper chamber. To initiate cell invasion, medium supplemented with 10% FBS was added as a chemoattractant in the low chamber. The cells were incubated at 37°C for 20 h and invaded cells on the inferior surface of the inserts were fixed, washed, and stained with crystal violet (Sigma-Aldrich). Each assay was repeated two times in duplicates. For analysis, the entire Matrigel surface area was imaged (Olympus SZX16 stereomicroscope, 2.5× magnification) and quantified using the ImageJ software. To ensure that the difference in invasion rate is not due to differential cell growth rate, total cell number validation was carried on a parallel plate. Cells were stained with crystal violet, lysed with 1% SDS, 0.1N NaOH solution followed by direct dye intensity measurement.

## Tumor initiation, histologic preparation, immunohistochemistry, and morphometric analysis

Subcutaneous tumors were generated by injecting $2 \times 10^6$ single-cell suspensions of cells in 100 µl PBS into a shaved lower right flank of 7-wk-old mice. D122 Control or *Bach1*-overexpressing cells were injected to male immunocompetent syngeneic C57BL/6 (Harlan Laboratories), whereas ES2 cells were injected to female immunodeficient CD1 nude (Harlan Laboratories) mice. Orthotopic metastatic tumor growth was initiated by injecting $1 \times 10^6$ ES2 cells

intraperitoneally at a remote site in the abdomen of 7-wk-old female CD1 nude mice (Harlan Laboratories). In all experiments, animals were randomly assigned to the control and overexpressing cells injected groups. Excised mouse tumors were rinsed in ice-cold PBS and then gradually fixed at 4°C in 2.5% and 1% PFA for 24 and 48 h, respectively. Paraffin-embedded tissue was sectioned serially at 4 $\mu$m thickness. The first slide was stained with hematoxylin and eosin, whereas other representative slides underwent immunohistochemical staining using the following antibodies: anti CD34 (1:100; Cedarlane Laboratories), LYVE1 (1:100; Fitzgerald Industries), cytokeratin 7 (ab9021, 1:200; Abcam), and VEGFC (H-190 1:200; Santa Cruz). Morphometric blood and lymphatic vessel coverage analysis were performed on CD-34 and LYVE1 stained tumor sections, respectively. Images of CD-34–stained sections were captured with a fluorescence microscope (NI-U; Nikon), equipped with Plan Fluor objectives connected to CCD camera (DS-Ri1; Nikon). Digital images were collected and assembled using Adobe Photoshop (Adobe Systems). LYVE1 immunohistochemical-stained sections were scanned using the Panoramic Viewer. The density of vessels was evaluated using Image Pro Plus software (Media Cybernetics). For all analysis, investigators were blinded and unaware of group allocation. All animal experiments described in this study were performed according to the guidelines of the Weizmann Institute Animal Care and Use Committee. For immunostaining of ES2 cells, fixation was carried out in 4% paraformaldehyde for 10 min. Fixed cells were then blocked for unspecific staining in 0.3% Triton X-100, 10% horse serum for 90 min at RT, incubated with antibodies directed either against HA-tag (1:200; Sigma-Aldrich) or VEGFC (H-190 1:200; Santa Cruz) antibody for 2 h at room temperature, and visualized by incubating the cells with goat anti-rabbit Cy3 (1:10,000; Jackson ImmunoResearch) secondary antibody. Nuclei were counterstained with DAPI (Invitrogen).

### Quantitative real-time-PCR

Total RNA was isolated from cultured cells or tumor specimens by PerfectPure RNA Cultured Cell or Tissue kit (5 PRIME), respectively, according to the manufacturer's instructions. RNA (1 $\mu$g) from each sample was subjected to first-strand cDNA synthesis with the High-Capacity cDNA Reverse Transcription kit (Applied Biosystems) and random hexamers. Quantitative real-time PCR was conducted with LightCycler-FastStart DNA Master SYBR Green I kit (Roche) using a LightCycler 480 real-time PCR System (Roche). The relative expression level of each target gene was determined using GAPDH and beta-2-microglobulin (B2M) as reference genes. Primers used were as follows: human *BACH1*: 5′-TCTTCCAGAAGAGGTGACAGT and 5′-ACTCCACA-CATTTGCACACT; *VEGFA*: 5′-ATGCGGATCAAACCTCACC and 5′-TCTTTCTTTGGTCTGCATTCAC; *VEGFB*: 5′-CCACCAGAGGAAAGTGGTGTC and 5′-ACAGCGCTGCACAGTCAC; *VEGFC*: 5′-GCCACGGCTTATGCAAGCAAAGAT and 5′-AGTTGAGGTTGGCCTGTTCTCTGT; *MMP1*: 5′-CTGGCCACAACTGCC-AAATG and 5′-CTGTCCCTGAACAGCCCAGTACTTA; *GAPDH*: 5′-AGGGCTG-CTT TTAACTCTGGT and 5′-CCCCACTTGATTTTGGAGGGA; *B2M*: 5′-TTCTGG-CCT GGAGGCTATC and 5′-TCAGGAAATTTGACTTTCCATTC; Mouse *Bach1*: 5′-TGACAGCGAGTCCTGTTCTG and 5′-TTATCCGTTGGGCATTGAA; *Vegfa*: 5′-TCT TCAAGCCATCCTGTGTG and 5′-GAGGTTTGATCCGCATAATCTG; *Vegfb*: 5′-ACGATGGCCTGGAATGTGTG and 5′-TGGTCTGCATTCACATTGGC; *Vegfc*: 5′-GTA-AAAACAAACTTTTCCCTAATTC and 5′-TTTAAGGAAGCACTTCTGTGTGT; *Gapdh*: 5′-GACGGCCGCATCTTCTTGTG and 5′-CTTCCCATTCTCGGCCTTGACTGT; *B2m*: 5′-CCCGCCTCACATTGAAATCC and 5′-GCGTATGTATCAGTCTCAGTGG; Zebrafish *bach2b*; 5′-CAGCATGCCAGAGGAGGT and 5′-AGTGATTGCTCTCCGACGC;

and *bactin* 5′-TGACAGGATGCAGAAGGAGA and 5′-GCCTCCGA TCCAGAC-AGAGT.

### Clearing and immunofluorescence staining

Excised tumors were fixed with 4% PFA for 1 wk at 4°C, washed three times with PBS, permeabilized with (0.2% Triton X-100 in PBS) for 4 h, and immersed overnight in blocking solution (PBS containing 0.05% Triton X-100 and 10% normal goat serum). For immunofluorescent staining, tumors were incubated with primary mouse anti-LYVE1 antibody (Fitzgerald Industries) diluted (1:250) in an antibody cocktail (50% blocking solution, 0.05% Triton X-100 in PBSX1) for 1 wk at 4°C, washed 24 h (1% blocking solution, 0.05% Triton X-100 in PBS), and probed with an Alexa Flour 594–conjugated goat antirabbit secondary antibody (A11037, 1:250; Molecular Probes—Life Technologies) diluted in antibody cocktail for an additional week at 4°C. After 24-h wash, the tumors were subjected to a Whole Organ Blood and Lymphatic Vessels Imaging (WOBLI) clearing procedure (Hama et al, 2011; Chung et al, 2013; Oren et al, 2018). Briefly, the tumors were re-fixed with 4% PFA for 24–72 h at 4°C, transferred to hydrogel solution (4% Acrylamide, 0.025% Bis-acrylamide, 0.25% Va-044 and 4% PFA in PBS) for 1 wk, and passively cleared (200 mM Boric acid and 4% SDS) for 2 wk at 37°C. Subsequently, the tumors were placed in Scale solution (4M urea, 10% glycerol and 0.1% Triton X-100) for 48 h. A sequence of 3D images and movies were acquired using Zeiss LSM710 confocal microscope (Carl Zeiss).

### Microscopy

Images were acquired on Zeiss LSM 780 upright confocal microscope (Carl Zeiss) with a W-Plan Apochromat ×20 objective, NA 1.0. Fluorescent proteins were excited with single-photon laser (488 nm). Alternatively, Leica TCS SP8 microscope, equipped with environmental control, two internal Hybrid (HyD) detectors, and Acusto Optical Tunable Filter (Leica microsystems CMS GmbH) was used, and excitation was performed using 488-nm Ar laser and emission was collected using the internal HyD detector at 510–625 nm, with a gain of 100.

### Statistical analysis

Statistical analyses were performed with analytic computerized software (Statistix 8 Student Edition, Analytical Software). Comparisons between treatment groups were performed with either ANOVA or Kruskal–Wallis nonparametric ANOVA (alternatively, t test or Wilcoxon rank sum test were used if only two groups were compared). ANOVA test was used to analyze normally distributed data (evaluated by Shapiro–Wilk test) that had equal variances between groups (evaluated by Bartlett's test), whereas Kruskal–Wallis nonparametric ANOVA was used to analyze data that were not normally distributed and/or had unequal variance between groups; when relevant, Tukey HSD test or mean ranks test, respectively, were used for all pairwise comparisons. Differences were considered significant at $P < 0.05$. Unless otherwise noted, data are presented as mean ± SEM.

## Supplementary Information

# Acknowledgements

The authors would like to thank Irit Orr and Ron Rotkopf for assistance with human cancer data analysis; Prof Eytan Domany and Dr Noa Ben-Moshe for transcription factor analysis; Ayala Sharp for FACS analysis support; Michal Shemesh for assisting with imaging requirements; Ofir Atrakchi, Yona Ely, Ron Hadas, Tal Lupo, and Gila Meir for technical assistance; Noa David for graphic design; Prof Lea Eisenbach for providing the mouse D122 Lewis lung carcinoma cell line; Gabriella Almog, Ala Glozman, and Roy Hofi for fish care; and Beni Siani and Alon Harmelin for animal care. This work was supported in part by the Israel Science Foundation grants 326/14 (to M Neeman and B Cohen), 748/2009 (to K Yaniv), and 861/2013 (to K Yaniv); European Research Council Advanced grant 232640-IMAGO (to M Neeman) and ERC Starting grant 335605 (to K Yaniv); the Thompson Foundation (to M Neeman); National Institutes of Health grant (R01 CA75334, to M Neeman); Marie Curie Actions-International Reintegration grants FP7-PEOPLE-2009-RG 256393 (to K Yaniv); the H&M Kimmel Institute for Stem Cell Research (to K Yaniv); and the Estate of Emile Mimran (SABRA program) (to K Yaniv). Michal Neeman is the incumbent of the Helen and Morris Mauerberger Chair in Biological Sciences. K Yaniv is supported by the Willner Family Center for Vascular Biology; the estate of Paul Ourieff; the Carolito Stiftung; Lois Rosen, Los Angeles, CA; Edith Frumin; the Fondazione Henry Krenter; the Wallach Hanna & Georges Lustgarten Fund; and the Polen Charitable Trust.

## Author Contributions

B Cohen: conceptualization, data curation, and writing—original draft.
H Tempelhof: data curation and investigation.
T Raz: formal analysis, validation, and writing—review and editing.
R Oren: data curation, formal analysis, methodology, and writing—review and editing.
J Nicenboim: data curation, investigation, and methodology.
F Bochner: data curation, investigation, visualization, and methodology.
R Even: data curation.
A Jelinski: data curation.
R Eilam: data curation, visualization, and methodology.
S Ben-Dor: formal analysis, methodology, and writing—review and editing.
Y Addadi: investigation and visualization.
O Golani: software, formal analysis, and visualization.
S Lazar: data curation and methodology.
K Yaniv: conceptualization, supervision, funding acquisition, project administration, and writing—review and editing.
M Neeman: conceptualization, funding acquisition, project administration, and writing—review and editing.

## Conflict of Interest Statement

The authors declare that they have no conflict of interest.

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
