## [Reviewer comments · Life Science Alliance]

Life Science Alliance

BACH family members regulate angiogenesis and lymphangiogenesis by modulating VEGFC expression

Batya Cohen, Hanoch Tempelhof, Tal Raz, Roni Oren, Julian Nicenboim, Filip Bochner, Ron Even, Adam Jelinski, Raya Eilam, Shifra Ben-Dor, Yoseph Addadi, Ofra Golani, Shlomi Lazar, Karina Yaniv, and Michal Neeman

DOI: <https://doi.org/10.26508/lsa.202000666>

Corresponding author(s): Michal Neeman, Weizmann Institute of Science and Karina Yaniv, Weizmann Institute of Science

Review Timeline:

Submission Date:	2020-02-02
Editorial Decision:	2020-02-03
Revision Received:	2020-02-18
Editorial Decision:	2020-02-19
Revision Received:	2020-02-23
Accepted:	2020-02-24

Scientific Editor: Andrea Leibfried

Transaction Report:

Please note that the manuscript was previously reviewed at another journal and the reports were taken into account in the decision-making process at Life Science Alliance.

Referee #1 Review

Report for Author:

This manuscript reveals the regulation of angiogenic and lymphangiogenic activity during zebrafish embryonic and tumor development by BACH family transcription factors. Based on an in silico screen the authors identify conserved BACH transcription factor binding sites in the human, mouse and zebrafish VEGFC promoters. Guided by expression in an endothelial reporter cell line Tg(fli1:EGFP), the function of the zebrafish paralogs BACH2a and BACH2b in vascular development was studied using antisense morpholinos (MO) and mutant fish generated by CRISPR/Cas9 genome editing. Processes analysed during zebrafish vascular development included formation of the primordial hindbrain channel (PHBC), formation of the parachordal cells

(PAC=lymphendothelial progenitors) and formation of the thoracic duct (TD). All three developmental processes revealed similar functions of BACH2a and 2b. (1.) MOs against BACH2a suppressed the developmental processes, while MOs against Bach2b were without effect, suggesting a predominant function of BACH2a. (2.) Unexpectedly, the same developmental processes were not impaired in BACH2a (-/-) mutants, suggesting a form of compensation. (3.) On a BACH2a (-/-) mutant background, MOs against BACH2b or CRISPR/Cas9 mediated destruction of BACH2b phenocopied the effect of MOs against BACH2a. This strongly suggested compensation of the loss of BACH2a by BACH2b. However, BACH2b mRNA upregulation was not the basis for this compensation. (4.) Lyve1 and VEGFC, but not FLT1 expression, were strongly reduced by BACH2a MOs. VEGFC MOs resulted in largely similar defects to BACH2a MOs and indeed provision of VEGFC mRNA rescued or at least ameliorated the defects caused by MOs against BACH2a, demonstrating that most of the defects caused by BACH2a MOs were likely caused by reduced / lacking VEGFC expression.

To investigate BACH transcription factors in tumor development, the authors forced expression of BACH1 in human ovarian clear cell carcinoma cells (ES2) and Lewis lung carcinoma cells (D122). After xenotransplantation, enhanced BACH1 expression in ES2 cells resulted in increased tumor blood and lymph vessel density and increased tumor cell invasion. D122 tumors showed increased VEGFC expression. Mechanistically, reporter constructs identified a mildly inhibitory proximal and more strongly stimulatory distal BACH-binding site in the VEGFC promoter of humans, mice and fish.

The data reported on the action of BACH family member in fish development and mammalian tumor formation have both merit. In particular, the zebrafish data are detailed and convincing, however, my major criticism is that the two parts of the manuscript, zebrafish development and tumor biology, are conceptually not well linked. Consequently, the study remains descriptive and does not provide deeper insights in either topic that are of broad general interest. From the developmental point of view the compensation mechanism between BACH2a and 2b in the fish is very interesting. Compensation is already active in heterozygous fish, which are refractory against BACH2a MO action, the authors exclude BACH2b mRNA upregulation as a possible mechanism, but do not further investigate or even only speculate about possible mechanisms. On the side of the tumor analysis also important questions remain open. Presence of tumor cells within lymphatic vessels is interpreted as indication of increased metastatic dissemination. However are these functional vessels? Spread via the lymphatic vessels could be analysed in the draining lymph nodes, spread via blood vessels through enumeration of e.g. lung metastasis. The tumor cell lines are amenable to CRISPR/Cas9 editing, would e.g. VEGFC deletion abrogate the increased pro-angiogenic effect of forced BACH expression, but leave the increased invasive activity untouched? VEGFC dependent and independent effect could also be distinguished by soluble VEGFR3-Fc.

Specific issues: Comment on the relationships (relative homologies) between the four fish BACH family members and the mammalian genes given that 2a and 2b are studied in the fish and BACH1 in the tumor context? Is there compensation for deletion of BACH 1 or 2 in mammalian cells, e.g. mRNA upregulation? How does BACH1 deletion or knock down in ES2 cells affect relative VEGFC expression? The initial choice to analyse BACH2a and 2b was based on expression of Tg(Fli1:EGFP)-positive cells. In the fish MOs and deletion of BACHs affect all cells, a large part of the phenotype is probably not caused by ECs (i.e. due to non-EC sources of VEGFC). Still MOs and deletion of BACHs affect ECs in a cell autonomous as well as a non-cell autonomous fashion, can the authors discuss this? Are BACH TF binding sites also present in the LYVE1 promoter? Forced BACH expression in tumor cells per definition would affect LECs only non-cell autonomously, which sets both part of the study mechanistically apart.

Minor issues:

Please provide page numbers

Fig.1 B: bactin, name beta actin in legend

Fig. 2 J: Why does the control MO cause a complete loss of normal blood flow?

Fig. 5 A: CD34 is expressed by tumor blood vessels but not exclusively. A higher magnification would be helpful in establishing that the signal is indeed exclusively vascular. In addition, PECAM staining can be used for verification.

Fig 5 C: Same argument, subsets of tissue macrophages express LYVE1, co-staining with Prox1 could be used to identify these cells. Please provide the z dimension for the volume reconstruction.

Fig. 5F. The histology of the BACH1 tumors is rather convoluted. Given the scale bars, the magnification should be roughly similar to Fig. 5C, which would suggest that majority of CK7-positive cells is located between lymph vessels rather in their lumen. Here are volume reconstruction like in Fig. 5 C could help to clearly identify vessel structures and cells within the vascular lumen.

Referee #2 Review

Report for Author:

The manuscript by Cohen et al. demonstrates a new mechanism of vascular endothelial growth factor (VEGF)-C regulation by the BTB and CNC homology (BACH) family of transcription factors, *bach2a* in the zebrafish and *Bach1* in mice. They used an elegant approach of identifying the BACH family of transcription factors as a putative modulator of VEGF-C expression from a human prostate cancer dataset. Their work in developmental context in zebrafish demonstrates impairment in normal blood and lymphatic vessel patterning, primarily centered around the *bach2a* isoform. They extend these observations by demonstrating human BACH1 interacts with a proximal site on the VEGFC promoter, resulting in a 23% increase in promoter activity, as well as a distal site, which results in an 86% reduction in promoter activity. Finally, the authors demonstrate BACH1 overexpression in human cancer cell lines increases VEGF-C, and that BACH1/VEGF-C levels correlate with human cancer progression (metastasis and tumor stage).

MAJOR COMMENTS:

1. The functional differences in BACH1/2 in humans (and mice) and *bach1a/b* and *bach2a/b* in zebrafish should be clarified in greater detail. In Fig. 6A, conservation of the consensus binding sequencing in human shows what appears to be BACH2 binding to the distal site (which appears to be an enhancer) whereas BACH1 interacts with the proximal site (which appears to be a repressor), based on the data in Fig. 6C, however, much of the subsequent work focuses on overexpression of BACH1 rather than BACH2, and it is unclear the rationale for this, as mutation of the putative BACH2 binding site results in impressive suppression of basal VEGF-C promoter activity and the developmental defects in angiogenesis and lymphangiogenesis in the zebrafish were shown as a result of *bach2a* targeting. Along these lines, it would also be of interest to determine whether stimuli relevant to the activation of VEGF-C in the context of the tumor microenvironment are also affected by BACH1/2, or if this is primarily a regulatory mechanism for basal VEGF-C expression. BACH1 has been reported to suppress angiogenesis dependent on its BTB domain (Jiang L et al. EBioMedicine. 2020) by the same group who demonstrated *Bach1* repression of Wnt/ β -catenin signaling (Jiang L et al. Oxid Med Cell Longev. 2017), although they do not mention interactions with the VEGF-C promoter or effects on lymphangiogenesis. This report is seemingly in conflict with the data presented in Fig. 7-S3A,C, where the authors demonstrate there is no significant difference in VEGF-A expression upon BACH1 overexpression, and this difference should be explained. The work in this paper also focuses on VEGF-C downstream of BACH1, and it is unclear the proposed

mechanism of VEGF-C-mediated regulation of angiogenesis. While VEGF-C can interact with VEGFR2 to induce angiogenesis *in vivo* (Cao Y et al. Proc Natl Acad Sci USA. 1998), it has a much lower affinity towards VEGFR2 than VEGFR3 (which is also expressed in vascular ECs).

2. It would be of interest to better understand why *bach2a* MO cause lymphatic defects only in *bach2a*mut^{+/+} zebrafish, whereas *bach2b* MO or gRNA only cause lymphatic defects in *bach2a*mut^{+/-} or *bach2a*mut^{-/-} embryos. The authors suggest a mechanism of genetic compensation in *bach2a*-Crispr mutants by *bach2b*, but show no difference in *bach2b* and actually report an increase in *bach2a* mRNA in homozygous mutants. Further, on page 15, the authors state that "knockout of *bach2a* expression (in mutants) triggers a functional compensation by *bach2b*..." despite showing no differences in *bach2b* expression between *bach2a*mut^{+/-} and *bach2a*mut^{-/-} embryos at 6dpf. Were the *bach2b* mRNA expression levels comparable at earlier time points, such as 30hpf or 3dpf, where some of the angiogenic and lymphangiogenic phenotypes were seen? Related to the comment #1, as it appears that BACH1 and BACH2 share the same consensus sequence, do they compete in some manner for interactions on the proximal and distal promoter sites, respectively? Considering the lack of observed compensation of *bach2b* mRNA, were levels of *bach1a* or *bach1b* altered in *bach2a* mutants?
3. While the authors demonstrate *bach2a* in Tg(*fli1*:EGFP)^{y1} GFP⁺ cells at 21-24hpf and 3dpf, the relative abundance in GFP⁻ cells appears higher. It is unclear whether the control of VEGF-C expression occurs in ECs/LECs, or rather another stromal cell type.
4. In the discussion, the authors highlight the possible heterodimerization between BACH and MAFB, however, it is unclear the nature of their possible interaction in the context of the current study. Were there any differences in PROX1, KLF4, NR2F2 or SOX18 in BACH mutants in LECs? The *mafba* and MAFB work was focused on LECs, whereas the current study investigates BACH1 in cancer cells, which may not have the same MAFB expression or interactions. It is unclear whether the authors are suggesting that BACH may also function to control MAFB signaling in LECs, contributing the lymphatic phenotype beyond the regulation of VEGF-C.
5. Could the authors comment on why *Bach1/2* knockout mice do not have developmental defects in angiogenesis and/or lymphangiogenesis?
6. The overall conclusion of the tumor studies would benefit from aligning the tumor implantation studies (ES2 human ovarian clear cell carcinoma cells and D122 Lewis lung carcinoma cells) with reports of BACH1 and VEGF-C expression in human cancers (human melanoma and human lung adenocarcinoma). The initial profiling for VEGF-C modulators was performed from a human prostate cancer dataset, begging the question whether BACH1 or VEGF-C are elevated in this cancer type. It would also be of interest to confirm BACH-mediated regulation of VEGF-C expression occurs in other cancer cell types, or whether this is unique to ES2 cells.
7. It would be of benefit to further discuss/differentiate the role of BACH1 in cancer, and the BACH1/VEGF-C-dependent pathway highlighted in this paper in the context of developmental angiogenesis/lymphangiogenesis. As noted in this manuscript, BACH1 has been reported to regulate other functions, including "oxidative stress, metabolism, cell transformation, neurodegenerative diseases, tumor expansion and metastatic spread", however, these BACH1 phenotypes have not been directly linked to VEGF-C. In the context of tumor studies, perhaps blocking VEGF-C signaling using an approach such as anti-VEGFR3 antibody to ameliorate the effects of BACH1 overexpression could serve to validate this point, with particular emphasis on angiogenic and lymphatic phenotypes.

MINOR COMMENTS:

1. It would be of interest to demonstrate *Bach1* mutations shown in Fig. 6D or knockdown/knockout of BACH1 result in a reduction of VEGF-C in ES2 cells
2. It appears as though error bars are missing on some graphs (Fig. 2E, Fig. 2-S2B,D,E,G-J, Fig. 2-S3A-C, Fig. 3D, Fig. 4D,E,G, Fig. 7-S2A-D).

3. The manuscript should be checked for typographical errors, as there are several instances of capitalization where unnecessary.

4. The manuscript should be checked for consistency of gene notation references as per species, where common convention dictates human genes should be listed with all lettering capitalized, whereas mouse genes should be listed with only the first letter capitalized and zebrafish genes all in lower case.

February 3, 2020

Re: Life Science Alliance manuscript #LSA-2020-00666-T

Michal Neeman
Weizmann Institute of Science
Dept. of Biological Regulation
Rehovot 76100 Israel
76100 Rehovot, N/A 76100
Israel

Dear Dr. Neeman,

Thank you for transferring your manuscript entitled "BACH family members regulate angiogenesis and lymphangiogenesis by modulating VEGFC expression" to Life Science Alliance. The manuscript was assessed by expert reviewers at another journal before, and the editors transferred those reports to us with your permission.

The reviewers appreciated your work, but would have expected more in-depth insight into the compensation mechanism between BACH2a and 2b and more insight into the role of the transcription factors in tumor development.

We concluded that the work is suitable for publication in Life Science Alliance without such further reaching insight. However, a point-by-point response to all concerns raised should get provided and the manuscript (text) modified accordingly. Please also note that we would need upload of all figures as individual files and that the supplementary figures should get numbered S1, S2,Sn.

Thank you for this interesting contribution to Life Science Alliance. We are looking forward to receiving your revised manuscript.

Sincerely,

Andrea Leibfried, PhD
Executive Editor
Life Science Alliance

Meyerhofstr. 1
69117 Heidelberg, Germany
t +49 6221 8891 502
e a.leibfried@life-science-alliance.org
www.life-science-alliance.org

B. MANUSCRIPT ORGANIZATION AND FORMATTING:

Point-by-point response to the reviewers' comments

Response to comments of Reviewer #1:

.....my major criticism is that the two parts of the manuscript, zebrafish development and tumor biology, are conceptually not well linked. Consequently, the study remains descriptive and does not provide deeper insights in either topic that are of broad general interest.

In light of the perception that the development of the early embryo shares many signaling pathways with cancer development and metastasis, we decided to evaluate the contribution of BACH family transcription factors to vascular development and remodeling during these two processes.

From the developmental point of view the compensation mechanism between BACH2a and 2b in the fish is very interesting. Compensation is already active in heterozygous fish, which are refractory against BACH2a MO action, the authors exclude BACH2b mRNA upregulation as a possible mechanisms, but do not further investigate or even only speculate about possible mechanisms.

We thank the reviewer for the comment. We have now addressed this by discussing the possible mechanisms that may provoke the compensation in the results.

Presence of tumor cells within lymphatic vessels is interpreted as indication of increased metastatic dissemination. However are these functional vessels? Spread via the lymphatic vessels could be analysed in the draining lymph nodes, spread via blood vessels through enumeration of e.g. lung metastasis. The tumor cell lines are amenable to CRISPR/Cas9 editing, would e.g. VEGFC deletion abrogate the increased pro-angiogenic effect of forced BACH expression, but leave the increased invasive activity untouched? VEGFC dependent and independent effect could also be distinguished by soluble VEGFR3-Fc.

Numerous reports suggest that increased expression of VEGFC in tumor correlates with lymphangiogenesis and lymph node metastasis in several malignancies. Moreover, it was demonstrated that in human or animal tumor models, tumor cells themselves can secrete high levels of VEGFC. This overexpression of tumor-derived VEGFC plays a crucial role in the occurrence of intratumoral-lymphangiogenesis leading to the dissemination of tumor cells to lymph vessels surrounding the tumor and thereafter to regional lymph nodes. Thus, we assume that the increased metastatic spread of tumor cells via the lymphatic vessels observed in BACH1-overexpressing transplanted ES2 tumor is due to increase in VEGFC expression.

Comment on the relationships (relative homologies) between the four fish BACH family members and the mammalian genes given that 2a and 2b are studied in the fish and BACH1 in the tumor context? Is there compensation for deletion of BACH 1 or 2 in mammalian cells, e.g. mRNA upregulation? How does BACH1 deletion or knock down in ES2 cells affect relative VEGFC expression?

According to our results (data not shown) and previously published reports (Luo Q. *et al.*, 2016; *Journal of Fish Biology* 88:1584–1597 and Fuse Y. *et al.*, 2015; *Genes to Cells* 20:590–600) *bach2a* and *2b* are more closely related to BACH2, and *bach1a* and *1b* to BACH1. It is clear that the two proteins have both sequence and functional differences. However, as transcription factors, they bind the same consensus sequence. Initially, we concentrated on BACH family members due to bioinformatics analysis which gives no indication which member is binding or active in any given biological context thus, we checked all possibilities. In fish, due to expression profile, we focused on *bach2* paralogs. In tumors, we decided to focus on BACH1 due to its higher expression in several published datasets and being identified in metastatic signatures in several studies.

Our recent preliminary results indicate that there is compensation in ES2 and D122 cells following BACH1 knockdown. We have found that VEGFC mRNA level was augmented in ES2/D122 BACH1-knockdown clones in comparison to control cells.

The initial choice to analyse BACH2a and 2b was based on expression of Tg(Fli1:EGFP)-positive cells. In the fish MOs and deletion of BACHs affect all cells, a large part of the phenotype is probably not caused by ECs (i.e. due to non-EC sources of VEGFC). Still MOs and deletion of BACHs affect ECs in a cell autonomous as well as a non-cell autonomous fashion, can the authors discuss this?

As the expression of *bach2a* and *vegfc* not overlap completely it supports context-specific and/or cell autonomous as well as a non-cell autonomous regulation of *vegfc* by *bach2*.

Are BACH TF binding sites also present in the LYVE1 promoter?

This question is beyond the context of this paper, however we checked the promoters of human, mouse and zebrafish LYVE1 (2600 bp upstream and 500 bp downstream of the transcription start site, as we did for VEGFC), and there is one conserved BACH binding site proximal to the transcription start site.

Minor issues

Please provide page numbers

Provided.

Fig. 1 B: bactin, name beta actin in legend

Modified.

Fig. 2 J: Why does the control MO cause a complete loss of normal blood flow?

You are correct, it doesn't, there was a mistake in the figure which was corrected.

Fig. 5 A: CD34 ist expressed by tumor blood vessels but not exclusively. A higher magnification would be helpful in establishing that the signal is indeed exclusively vascular. In addition, PECAM staining can be used for verification.

The image was replaced by a higher magnification image showing that the CD34 signal is exclusively vascular.

Fig 5 C: Same argument, subsets of tissue macrophages express LYVE1, co-staining with Prox1 could be used to identify these cells. Please provide the z dimension for the volume reconstruction.

As requested, Z dimensions were added to the legends of Figure 5, movie 1 and 2.

Fig. 5F. The histology of the BACH1 tumors is rather convoluted. Given the scale bars, the magnification should be roughly similar to Fig. 5C, which would suggest that majority of CK7-positive cells is located between lymph vessels rather in their lumen. Here are volume reconstruction like in Fig. 5 C could help to clearly identify vessel structures and cells within the vascular lumen.

The immunofluorescence staining presented in Figure 5F was carried out on 4µm thickness specimen sectioned from paraffin-embedded diaphragm. This information is now included in the figure legend.

Response to comments of Reviewer #2:

Major comments

1. The functional differences in BACH1/2 in humans (and mice) and bach1a/b and bach2a/b in zebrafish should be clarified in greater detail. In Fig. 6A, conservation of the consensus binding sequencing in human shows what appears to be BACH2 binding to the distal site (which appears to be an enhancer) whereas BACH1 interacts with the proximal site (which appears to be a repressor), based on the data in Fig. 6C, however, much of the subsequent work focuses on overexpression of BACH1 rather than BACH2, and it is unclear the rationale for this, as mutation of the putative BACH2 binding site results in impressive suppression of basal VEGF-C promoter activity and the developmental defects in angiogenesis and lymphangiogenesis in the zebrafish were shown as a result of bach2a targeting. Along these lines, it would also be of interest to determine whether stimuli relevant to the activation of VEGF-C in the context of the tumor microenvironment are also affected by BACH1/2, or if this is primarily a regulatory mechanism for basal VEGF-C expression. BACH1 has been reported to suppress angiogenesis dependent on its BTB domain (Jiang L et al. EBioMedicine. 2020) by the same group who demonstrated Bach1 repression of Wnt/β-catenin signaling (Jiang L et al. Oxid Med Cell Longev. 2017), although they do not mention interactions with the VEGF-C promoter or effects on lymphangiogenesis. This report is seemingly in conflict with the data presented in Fig. 7-S3A,C, where the authors demonstrate there is no significant difference in VEGF-A expression upon BACH1 overexpression, and this difference should be explained. The work in this paper also focuses on VEGF-C downstream of BACH1, and it is unclear the proposed mechanism of VEGF-C-mediated regulation of angiogenesis. While VEGF-C can interact with VEGFR2 to induce angiogenesis in vivo (Cao Y et al. Proc Natl Acad Sci USA. 1998), it has a much lower affinity towards VEGFR2 than VEGFR3 (which is also expressed in vascular ECs).

In mice and humans BACH1 and BACH2 share the same consensus sequence. While the databases bioinformatically define "BACH1" and "BACH2" (as was presented in Fig1 and Fig 6) the consensus sequences are extremely similar, and it is not clear which factor will bind in a

specific biological context. However, BACH1 is expressed relatively ubiquitously playing a crucial role during tumor expansion, while BACH2 is predominantly expressed in B and T lymphocytes. As this paper is concentrating on the functional relationship between BACH family members and VEGFC it was relevant to assess the role of BACH1 during tumor progression and not that of BACH2. Conversely, *bach2* genes were studied during zebrafish development as their expression was more abundant in *Tg(Fli1:EGFP)*-positive and *Tg(Fli1:EGFP)*-negative cells than *bach1b*, while *bach1a* is barely detectable in the GFP-positive (GFP⁺) cell population, at either 21-24 hpf or 3 dpf (Fig 1B). The result concerning VEGFA expression may not be in conflict with any of the reports showing that VEGFA is suppressed by Bach1 as the other analyses were conducted in endothelial cells while in this study evaluation was carried out in tumor cells.

2. It would be of interest to better understand why bach2a MO cause lymphatic defects only in bach2amut+/+ zebrafish, whereas bach2b MO or gRNA only cause lymphatic defects in bach2amut+/- or bach2amut/- embryos. The authors suggest a mechanism of genetic compensation in bach2a-Crispr mutants by bach2b, but show no difference in bach2b and actually report an increase in bach2a mRNA in homozygous mutants. Further, on page 15, the authors state that "knockout of bach2a expression (in mutants) triggers a functional compensation by bach2b..." despite showing no differences in bach2b expression between bach2amut+/- and bach2amut/- embryos at 6dpf. Were the bach2b mRNA expression levels comparable at earlier time points, such as 30hpf or 3dpf, where some of the angiogenic and lymphangiogenic phenotypes were seen? Related to the comment #1, as it appears that BACH1 and BACH2 share the same consensus sequence, do they compete in some manner for interactions on the proximal and distal promoter sites, respectively? Considering the lack of observed compensation of bach2b mRNA, were levels of bach1a or bach1b altered in bach2a mutants?

All of our results point to compensation post-transcriptionally. Unfortunately, antibodies are not available to test the protein levels or DNA binding for zebrafish, and the antibodies available for mammalian BACH family members do not recognize the zebrafish proteins (data not shown). As the compensatory mechanism is atypical, it is beyond the scope of this paper, which concentrates on the regulatory relationship between BACH and VEGFC. Indeed, we do not know which of the BACH family members bind the consensus sites, and may compete for binding.

3. While the authors demonstrate bach2a in Tg(fli1:EGFP)y1 GFP+ cells at 21-24hpf and 3dpf, the relative abundance in GFP- cells appears higher. It is unclear whether the control of VEGF-C expression occurs in ECs/LECs, or rather another stromal cell type.

The lack of full overlapping expression between *bach2a* and *vegfc* supports additional tissue-specific functions for each of these factors.

4. In the discussion, the authors highlight the possible heterodimerization between BACH and MAFB, however, it is unclear the nature of their possible interaction in the context of the current study. Were there any differences in PROX1, KLF4, NR2F2 or SOX18 in BACH mutants in LECs? The mafba and MAFB work was focused on LECs, whereas the current study investigates BACH1 in cancer cells, which may not have the same MAFB expression or interactions. It is unclear whether the authors are suggesting that BACH may also function to control MAFB signaling in LECs, contributing the lymphatic phenotype beyond the regulation

of VEGF-C.

The minor discussion point was raised as BACH belongs to the MAF family, and there are known interactions between the various members. In addition, the fact that VEGFC affects MAF family members, and here we show that BACH effects VEGFC, we felt that it was relevant to point out that they are related.

5. Could the authors comment on why *Bach1/2* knockout mice do not have developmental defects in angiogenesis and/or lymphangiogenesis?

We predict that in mice, similar to the data shown here, *Bach1* and *Bach2* may work in a yet unknown compensatory mechanism. As previously reported *Bach1*^{-/-} (Sun J. *et al.*, *EMBO J* 2002; 21: 5216–24 or *Bach2*^{-/-} (Muto A. *et al.*, 2004; *Nature* 429:566–571) progeny displayed no obvious abnormalities. However, specific developmental defects in angiogenesis and/or lymphangiogenesis may not have been investigated.

6. The overall conclusion of the tumor studies would benefit from aligning the tumor implantation studies (ES2 human ovarian clear cell carcinoma cells and D122 Lewis lung carcinoma cells) with reports of BACH1 and VEGF-C expression in human cancers (human melanoma and human lung adenocarcinoma). The initial profiling for VEGF-C modulators was performed from a human prostate cancer dataset, begging the question whether BACH1 or VEGF-C are elevated in this cancer type. It would also be of interest to confirm BACH-mediated regulation of VEGF-C expression occurs in other cancer cell types, or whether this is unique to ES2 cells.

We show in the manuscript that VEGFC expression was specifically elevated in BACH1-overexpressing ES2 (ovarian) and D122 (lung) tumors both at the protein (Figs 7H, and Fig S7J, respectively) and mRNA level (Figs 7I and Fig S7I). We intend to study additional tumor cells including prostate cancer cells.

7. It would be of benefit to further discuss/differentiate the role of BACH1 in cancer, and the BACH1/VEGF-C-dependent pathway highlighted in this paper in the context of developmental angiogenesis/lymphangiogenesis. As noted in this manuscript, BACH1 has been reported to regulate other functions, including "oxidative stress, metabolism, cell transformation, neurodegenerative diseases, tumor expansion and metastatic spread", however, these BACH1 phenotypes have not been directly linked to VEGF-C. In the context of tumor studies, perhaps blocking VEGF-C signaling using an approach such as anti-VEGFR3 antibody to ameliorate the effects of BACH1 overexpression could serve to validate this point, with particular emphasis on angiogenic and lymphatic phenotypes.

We thank the reviewer for the suggestion. We intend to study the role of BACH1/VEGFC signaling axis in the context of oxidative stress and metabolism during tumor expansion.

Minor Comments

1. It would be of interest to demonstrate *Bach1* mutations shown in Fig. 6D or knockdown/knockout of BACH1 result in a reduction of VEGF-C in ES2 cells

As Fig 6 was a luciferase assay, there is no way with these constructs to check VEGFC levels. Our preliminary results indicate that upon BACH1 knockdown in ES2 and D122 cells following

VEGFC mRNA level was augmented in the majority of ES2/D122 BACH1-knockdown clones in comparison to control cells.

2. It appears as though error bars are missing on some graphs (Fig. 2E, Fig. 2-S2B,D,E,G-J, Fig. 2-S3A-C, Fig. 3D, Fig. 4D,E,G, Fig. 7-S2A-D).

All graphs were revised.

3. The manuscript should be checked for typographical errors, as there are several instances of capitalization where unnecessary.

Corrected.

4. The manuscript should be checked for consistency of gene notation references as per species, where common convention dictates human genes should be listed with all lettering capitalized, whereas mouse genes should be listed with only the first letter capitalized and zebrafish genes all in lower case.

The manuscript was reviewed and corrected accordingly.

February 19, 2020

RE: Life Science Alliance Manuscript #LSA-2020-00666-TR

Prof. Michal Neeman
Weizmann Institute of Science
Dept. of Biological Regulation
Rehovot 76100 Israel
76100 Rehovot, N/A 76100
Israel

Dear Dr. Neeman,

Thank you for submitting your revised manuscript entitled "BACH family members regulate angiogenesis and lymphangiogenesis by modulating VEGFC expression". I appreciate your point-by-point response and the introduced changes and I would thus be happy to accept your paper for publication here, pending final minor revisions necessary to meet our formatting guidelines:

- please list 10 authors et al. in your reference list
- I appreciate the section about statistical analysis, but please also mention which statistical test has been used in the figure legends (next to the p-value)

A. FINAL FILES:

-- Summary blurb (enter in submission system): A short text summarizing in a single sentence the study (max. 200 characters including spaces). This text is used in conjunction with the titles of papers, hence should be informative and complementary to the title. It should describe the context

and significance of the findings for a general readership; it should be written in the present tense and refer to the work in the third person. Author names should not be mentioned.

B. MANUSCRIPT ORGANIZATION AND FORMATTING:

Sincerely,

Andrea Leibfried, PhD
Executive Editor
Life Science Alliance
Meyerohofstr. 1
69117 Heidelberg, Germany
t +49 6221 8891 502
e a.leibfried@life-science-alliance.org
www.life-science-alliance.org

February 24, 2020

RE: Life Science Alliance Manuscript #LSA-2020-00666-TRR

Prof. Michal Neeman
Weizmann Institute of Science
Dept. of Biological Regulation
Rehovot 76100 Israel
76100 Rehovot, N/A 76100
Israel

Dear Dr. Neeman,

Thank you for submitting your Research Article entitled "BACH family members regulate angiogenesis and lymphangiogenesis by modulating VEGFC expression". It is a pleasure to let you know that your manuscript is now accepted for publication in Life Science Alliance. Congratulations on this interesting work.

DISTRIBUTION OF MATERIALS:

Again, congratulations on a very nice paper. I hope you found the review process to be constructive and are pleased with how the manuscript was handled editorially. We look forward to future exciting submissions from your lab.

Sincerely,

Andrea Leibfried, PhD
Executive Editor
Life Science Alliance
Meyerohofstr. 1
69117 Heidelberg, Germany
t +49 6221 8891 502
e a.leibfried@life-science-alliance.org
www.life-science-alliance.org